# WRKY Transcription Factors in Cassava Contribute to Regulation of Tolerance and Susceptibility to Cassava Mosaic Disease through Stress Responses

**DOI:** 10.3390/v13091820

**Published:** 2021-09-13

**Authors:** Warren Freeborough, Nikki Gentle, Marie E.C. Rey

**Affiliations:** School of Molecular and Cell Biology, University of Witwatersrand, Johannesburg 2000, South Africa; w.freeborough9@gmail.com (W.F.); nikki.gentle@wits.ac.za (N.G.)

**Keywords:** geminivirus, South African cassava mosaic virus, WRKY transcription factors, phytohormones, cassava mosaic disease, tolerance, susceptibility

## Abstract

Among the numerous biological constraints that hinder cassava (*Manihot esculenta* Crantz) production, foremost is cassava mosaic disease (CMD) caused by virus members of the family *Geminiviridae*, genus *Begomovirus*. The mechanisms of CMD tolerance and susceptibility are not fully understood; however, CMD susceptible T200 and tolerant TME3 cassava landraces have been shown to exhibit different large-scale transcriptional reprogramming in response to South African cassava mosaic virus (SACMV). Recent identification of 85 MeWRKY transcription factors in cassava demonstrated high orthology with those in *Arabidopsis*, however, little is known about their roles in virus responses in this non-model crop. Significant differences in *MeWRKY* expression and regulatory networks between the T200 and TME3 landraces were demonstrated. Overall, WRKY expression and associated hormone and enriched biological processes in both landraces reflect oxidative and other biotic stress responses to SACMV. Notably, *MeWRKY11* and *MeWRKY81* were uniquely up and downregulated at 12 and 67 days post infection (dpi) respectively in TME3, implicating a role in tolerance and symptom recovery. AtWRKY28 and AtWRKY40 homologs of MeWRKY81 and MeWRKY11, respectively, have been shown to be involved in regulation of jasmonic and salicylic acid signaling in *Arabidopsis*. AtWRKY28 is an interactor in the RPW8-NBS resistance (R) protein network and downregulation of its homolog *MeWRKY81* at 67 dpi in TME3 suggests a negative role for this WRKY in SACMV tolerance. In contrast, in T200, nine *MeWRKY*s were differentially expressed from early (12 dpi), middle (32 dpi) to late (67 dpi) infection. *MeWRKY27* (homolog *AtWRKY*33) and *MeWRKY55* (homolog *AtWRKY53*) were uniquely up-regulated at 12, 32 and 67 dpi in T200. AtWRKY33 and AtWRKY53 are positive regulators of leaf senescence and oxidative stress in *Arabidopsis*, suggesting MeWRKY55 and 27 contribute to susceptibility in T200.

## 1. Introduction

Cassava (*Manihot esculenta* Crantz) is a perennial food security crop largely grown in developing countries, with worldwide cassava productions measuring over 300 million tonnes in 2019 (http://www.fao.org/faostat/en/#data). Cassava’s food security crop status is attributed to its natural drought tolerance, which makes it a desirable crop for subsistence farmers [1]. However, cassava in sub-Saharan Africa and Asia is currently under threat from several pests and diseases, one of the most important being cassava mosaic disease (CMD) [2,3]. CMD presents as a spectrum of symptoms such as leaf curl, stunted growth, mosaic and chlorosis [4]. This subsequently leads to a reduction in cassava crop production, which can be as high as a 100% loss in yield [2]. Several species and genetic variants of begomoviruses are responsible for CMD, and these are classified under the family *Geminiviridae*, genus *Begomovirus* [5]. Cassava mosaic begomoviruses are bipartite, composed of two circular ssDNA components (DNA A and B) that are packaged separately into twinned geminate virion particles [6,7]. Each DNA component ranges between 2.5 and 2.9 kb in length and collectively codes for all the proteins needed for replication, cellular movement, vector transmission and the establishment of infection in the host [6,7]. *South African cassava mosaic virus* is one of the species causing CMD [8] and occurs in South Africa, Madagascar and Mozambique [9]. South African cassava mosaic virus (SACMV) and other begomoviruses are transmitted by the whitefly vector, (*Bemisia tabaci* (Genn.) [10].

The plant immune system comprises two interconnected mechanisms, namely pathogen triggered immunity (PTI) and effector-triggered immunity (ETI) [11]. PTI acts primarily in basal defense of the plants against infection, using pathogen recognition receptors (PRR) on the cells membrane surface to detect and respond to conserved pathogen molecules [12]. ETI functions predominantly within the cell through nucleotide binding-leucine rich repeat (NB-LRR) resistance proteins [13] which recognize viral effectors, initiating downstream signaling and defense responses, such as PR proteins [14,15] amongst others [16]. Systemic acquired resistance (SAR) and induced systemic resistance (ISR) are two forms of induced resistance wherein plant defenses are preconditioned by prior infection or treatment that results in resistance against subsequent challenge by a pathogen [17,18]. Additionally, plants respond to general biotic stresses, whose pathways are inextricably linked in a broad network of molecular interactions associated with defense responses. Plants activate biotic stress responses to pathogen invasion and activate appropriate regulatory or transcriptional pathways. Plant defense against pathogen attack is well understood, but the interplay and impact of different signals to generate defense responses against biotic stress still remain elusive. Most pathogens trigger a complex interconnected plant signaling network, suggesting that differences between SAR, PTI and ETI in different host-virus interactions are both quantitative and temporal as a result of fine-tuning and cross-talk between regulatory networks [19]. Transcriptional reprogramming is a prominent feature of plant immunity or susceptibility and is governed by transcription factors (TFs) and co-regulatory factors associated with distinct transcriptional complexes [20,21]. Different hormone signaling pathways converge on TFs forming sensory TF regulatory networks [16]. Phytohormones are used by plants to respond to a diverse range of stimuli, including biotic stress from pathogens and general stress responses. Hormones downstream of pathogen detection provide further regulation [16] and also form networks in plant immunity [22]. In particular, the most central hormones associated with biotic stress are ethylene (ET), abscisic acid (ABA), nitric oxide (NO), jasmonic acid (JA) and salicylic acid (SA). Traditionally, JA and ET modulate necrotrophic pathogens, while biotrophic and hemi-biotrophic pathogens respond to SA [23]. Furthermore, ABA is shown to be involved in the mediation of JA-SA interactions. For example, AtMYC2 functions both as a positive regulator for ABA signaling and the central regulatory hub in JA signaling [24,25]. JA and SA are considered the most important hormones as all other phytohormones modulate plant immunity through interacting either with JA or SA, which exist in an antagonistic relationship [22]. Hormones function both upstream and downstream of TFs. Hormones may alter gene expression by transducing the signal downstream through various MAPK cascades and phosphatases that induce TFs to initiate transcriptomic reprogramming and activate stress responses [26]. Conversely, the TFs may alter downstream hormone levels through the expression of target genes that either inhibit or induce hormone biosynthesis. Consequently, these networks of cross-talk between hormones and TFs enable the plant to respond adequately and flexibly to stress. These networks form an integral part of the plant and form the basis of regulation of plant functions [27]. Viruses optimize their strategy of reprogramming the host system by targeting hub proteins in signal transductions pathways [27]. These hub proteins are a network motif describing a node that is disproportionally interconnected compared to neighboring nodes, and through targeting hubs, viruses increase the number of induced perturbations in the plant hosts [28]. Transcription factors responsible for large scale transcriptional reprogramming principally belong to the ERF, NAC, MYB, bZIP and WRKY protein families [29]. Of these TFs, the WRKY family, in particular, play a pivotal role in regulating plant defenses [30,31,32,33,34,35]. WRKY proteins contain the conserved DNA-binding domain, comprising the WRKYGQK peptide and zinc-finger-like motifs, that binds to the W-box (TTGAC(C/T)) cis-regulatory element of target genes [31]. The WRKY family is a large family of TFs found in higher plants [34] and is implicated in several physiological processes such as seed development, dormancy and senescence [33]. WRKYs were previously regarded as plant-specific, but their identification in lower organisms such as a slime mold and protists, implies an earlier origin [35]. Despite this, most recently, the link between animals and plant WRKYs has been found. Interestingly, NF-κB in mice, a TF with functions in basal immunity, was found to bind analogous WT-boxes, similar to WRKYs studied in *Arabidopsis* [36]. Three broad subgroups exist in the WRKY TF family. Group I is characterized by having two WRKY domains and two C2H2 zinc finger domains. Group II and III both contain a single WRKY domain but contain different zinc finger domains (C2H2 and C2HC respectively) [32]. Group II is further divided into II a, b, c, d and e based on their amino acid sequence or phylogenetic data [35]. WRKYs were first shown to respond to pathogen infection in parsley, functioning as both positive and negative regulators of the pathogen response in plants via PTI and ETI [37]. Since then, the number of identified WRKYs has increased in multiple plant species, with the fewest being found in barley (*Hordeum vulgare*) [38], while soybean (*Glycine max)* has the highest number (182) characterized [39]. More recently, 85 *Manihot esculenta* (*Me*) *WRKY*s were characterized in cassava [40], 11 more than *AtWRKY*s found in *Arabidopsis thaliana* [34]. It has been shown that WRKYs are involved in innate immunity, defense gene activation and response to biotic stress [41]. A recent study reported that MeWRKY75 positively regulates disease resistance against cassava bacterial blight (*Xanthomonas axonopodis* pv. manihotis) [42]. While several studies have been performed on the roles of WRKYs in bacterial and fungal pathogens [43,44,45,46,47,48,49,50,51,52], very few virus-responsive WRKYs have been reported. WRKY8 has been shown to play a role in defense to crucifer-infecting tobacco mosaic virus (TMV-cg) in *Arabidopsis* by mediating both ABA and ET signaling [53]. For example, WRKY18, WRKY40 and WRKY60 repress expression of two ABA-responsive genes *ABI4* and *ABI5* in *Arabidopsis* [54]. Modulating the function of transcription factors through interactions with regulatory proteins is a critical process in the repression or activation of signal transduction pathways [49,55]. WRKY regulatory proteins form a complex interconnected regulatory network with themselves, and other TFs and interacting regulatory partners, such as Mitogen-Activated Protein kinases (MAPKs) and the key SA regulator, NPR1, amongst others [56,57,58]. Furthermore, numerous members of the WRKY family interact with various hormones required in signaling defense responses [33,44]. More recently, small interfering (siRNAs) and microRNAs (miRNAs) have also emerged as key regulators of WRKY networks via posttranscriptional gene silencing (PTGS) [59].

A previous study was performed quantifying the global transcriptomic response in the CMD-tolerant TME3 and susceptible T200 cassava landraces [60]. SACMV-induced disease progression was monitored over the early asymptomatic (12 dpi), middle symptomatic (32 dpi) and late recovery (67 dpi) infection stages. Persistent lower virus titers in TME3 compared with T200 were recorded [60,61]. Amongst the transcriptome results [60] several TFs were found to be differentially expressed in T200 and TME3 and provided the basis for a thorough functional investigation and comparison of *MeWRKY* TF expression in the two cassava landraces in this study. Differentially expressed *MeWRKY*s in T200 and TME3 at the three time points post SACMV infection were identified and mapped to *Arabidopsis* homolog networks where interacting partners were also revealed. Protein–protein interaction and enriched GO Biological Process results herein demonstrated, for the first time, significant differences in WRKY regulatory networks between a CMD-susceptible and tolerant cassava landrace. Notably, many *MeWRKY* expression changes were associated with hormone and biotic stress responses. *MeWRKY81* was uniquely downregulated at 67 dpi in TME3, suggesting a role in tolerance and symptom recovery. *MeWRKY3, MeWRKY27* and *MeWRKY*55 were up-regulated throughout infection (at 12, 32 and 67 dpi) in T200, implicating these TFs in severe disease development in this susceptible landrace. This is the first detailed study of *WRKY* TF responses to CMD in cassava.

## 2. Materials and Methods

### 2.1. Sample Collection

T200 is a susceptible southern African landrace, while TME3 is a CMD-tolerant West African landrace. The cassava landraces, TME3 and T200, were micro-propagated by nodal cutting culture, plantlets were agro-inoculated, and systemic infection of the cassava leaf tissue with *South African cassava mosaic virus* was confirmed by conventional PCR, as described by [60]. Briefly, newly developed symptomatic leaf tissue from apical leaves was collected from both SACMV-infected plants and mock-inoculated controls, from both landraces, at three time points, namely 12, 32 and 67 days post infection (dpi). These three time points represent early, middle and late stages of SACMV infection, respectively. In each case, two or three leaves under the apex were selected and pooled from six plants (per biological replicate). Three biological replicates were undertaken for SACMV-infected and mock-inoculated plants from both T200 and TME3 at 12, 32 and 67 dpi) [60].

### 2.2. Library Preparation and Sequencing

Total RNA was extracted from SACMV-infected and mock-inoculated leaf tissue at each of the three time points, using a modified high molecular weight polyethylene glycol (HMW-PEG) protocol [62], after which RNA concentrations were determined using the NanoDrop™ 1000 spectrophotometer (Thermo Fisher Scientific, Waltham, MA, USA) and the RNA integrity of the samples was assessed using an Agilent 2100 Bioanalyzer (Agilent, Santa Clara, CA, USA). Sequencing libraries were prepared from 12 μg of total RNA at the Functional Genomics Center UNI ETH Zurich, Switzerland, using the SOLiD Total RNA-Seq Kit (Applied Biosystems, Foster City, CA, USA), according to the manufacturer’s instructions. The 12 resulting libraries were multiplexed using the SOLiD RNA Barcoding Kit (Applied Biosystems) and pooled in an equimolar ratio. Paired-end sequencing (50 bp in the forward- and 35 bp in the reverse direction) was performed using the ABI SOLiD V4 system. An average of 11.42 million sequencing reads was obtained per sample. The raw sequencing data are available in the NCBI Sequence Read Archive (SRA), under the accession number, PRJNA255198. The sequencing data was verified by RT-qPCR, including WRKY70 [60]. The standard curve method was used to determine expression values for each target gene. Log2 gene expression patterns (up or downregulated) were in agreement with those observed in SOLiD sequencing data.

### 2.3. Pseudoalignment, Transcript Quantification and Differential Gene Expression Analysis

Sequencing resulted in two csfasta and two quality files per sample, which were then converted into csfastq format using the solid2fastq command, implemented through MAQ v0.6.6. The resulting FASTQ files were then pseudoaligned to the *Manihot esculenta* v6.1 genome (https://phytozome.jgi.doe.gov/Mesculenta, accessed on 3 November 2019) using Salmon v0.14.1 [63], to obtain transcript-level quantifications of gene expression. Tximport v1.12.3 [64] was used to aggregate the transcript quantifications to the gene level, in preparation for differential gene expression analysis. Genes differentially expressed between the SACMV-infected plants and mock-inoculated controls in both TME3 and T200 were then identified at each of the three time points using DESeq2 v1.24.0. [65]. A significance threshold of α = 0.05 was used to identify statistically significantly differentially expressed genes. Only statistically significantly differentially expressed genes with an absolute log_2_ fold change in expression greater than 1.5 were included in subsequent analyses.

### 2.4. Annotation of M. esculenta Genes Differentially Expressed in TME3 and T200, at 12, 32 and 67 dpi

Statistically significantly differentially expressed *M. esculenta* genes with an absolute log_2_ fold change in expression greater than 1.5, in both TME3 and T200, at all three time points, were annotated with information relating to their *Arabidopsis thaliana* homologs, based on their The Arabidopsis Information Resource (TAIR; https://www.arabidopsis.org, accessed on 3 November 2019) gene identifiers, and associated gene ontology (GO) terms and PFAM (http://pfam.xfam.org, accessed on 3 November 2019) domains. Annotation was performed using the BioMart tool available through Ensemble Plants (release 42; https://jul2018-plants.ensembl.org, accessed on 3 November 2019), against the Ensembl Plants Genes 40 database and Manihot_esculenta_v6 dataset.

### 2.5. Identification of MeWRKYs Differentially Expressed in TME3 and T200, at 12, 32 and 67 dpi, and Their Interacting Partners

Following annotation of the differentially expressed genes, Me*WKRYs* differentially expressed at 12, 32 and 67 dpi, in both TME3 and T200, were identified based on their presence on the list of 85 MeWRKYs identified by Wei et al. (2016) [40]. Each of these MeWRKYs was then separately queried against the STRING v11.1 [66], in order to identify their interacting partners, based on protein–protein interaction networks built using data derived from *A. thaliana*. The minimum required interaction score was set to a custom value of 0.600, where the first shell included only the queried *Arabidopsis* homologs, and the second shell was set to include no more than 50 interactors. The resulting list of interacting partners was then used to construct landrace- and time point-specific networks centered around the WRKYs, where differentially expressed genes, within each landrace, at each time point, were colored to indicate up- and downregulation. All networks were visualized and annotated using Cytoscape v3.5.1 [67].

### 2.6. Identification of GO Terms Significantly Enriched within M. esculenta Genes Differentially Expressed in TME3 and T200, at 12, 32 and 67 dpi

For both the TME3 and T200 cassava landraces, at 12, 32 and 67 dpi, the statistically significantly differentially expressed genes with an absolute log_2_ fold change in expression greater than 1.5 were further classified as being either up- or downregulated. Up- and downregulated gene sets from each landrace, at each time point, were then assessed in order to identify particular biological processes or molecular functions enriched within these gene sets, relative to the total set of expressed *A. thaliana* transcripts (obtained from TAIR10). This GO enrichment analysis was performed using clusterProfiler v2.7 [68], implemented in R v3.3.1. Correction for multiple testing using the Benjamini-Hochberg false discovery rate was applied, such that GO terms in each gene set with an adjusted q-value < 0.05 were classified as significantly enriched.

### 2.7. Identification of M. esculenta Genes Differentially Expressed in TME3 and T200, at 12, 32 and 67 dpi Significantly Enriched within Hormone Signaling-Related Pathways

*M. esculenta* genes within GO terms significantly enriched in TME3 and T200, at 12, 32 and 67 dpi, associated with the abscisic acid (ABA; GO:0009738 and GO:0071215), salicylic acid (SA GO:0009751, GO:0009863, GO:0071446, GO:0009862, GO:0009697, GO:0009696) jasmonic acid (JA, GO:0009753, GO:0009867, GO:0071395, GO:0009694, GO:0009695 and GO:0009861) and ethylene (ET, GO:0009692, GO:0009693, GO:0009723, GO:0009873, GO:0071369, GO:0009861, GO:0010105 and GO:0010104) hormone signaling pathways were identified using clusterProfiler v2.7 [68], implemented in R v3.3.1. Genes within these gene sets were then further identified as being either up- or downregulated based on comparison with the corresponding sets of statistically significantly differentially expressed genes with an absolute log_2_ fold change in expression greater than 1.5 from each landrace, at each time point.

The total number of genes within each hormone signaling pathway that were up- or downregulated (within each landrace, at each time point) was then counted. Genes associated with more than one GO term ABA, SA, JA, and ET signaling were only counted once, in order to avoid overrepresentation. After removing duplicate entries, the proportion of the total set of genes in each hormone-related GO term represented by these genes was then determined. The biological networks between these genes were then visualized and annotated using Cytoscape v3.5.1 [67].

### 2.8. Quantitative PCR of Selected WRKYs

Quantitative PCR was performed on *MeWRKY3, 11 and 18/59* to validate expression of the transcriptome results (Appendix A). Total RNA was extracted from previously sampled leaf tissue using QIAzol (Qiagen, Germantown, MD, USA) following the manufacturer’s instructions. RNA was purified using RNA clean-up and concentrator (Zymo Research). RNA purity was analysed on a spectrophotometer (model) A280/A280 > 1.8 and A260/230 > 1.8. Complementary DNA synthesis was performed using RNA as a template, following the instructions prescribed for the RevertAid H Minus First Strand cDNA Synthesis Kit (ThermoFisher Scientific, Waltham, MA, USA). Reverse transcriptase qPCR relative gene expression was performed in triplicate for each gene, with cDNA as template using the Maxima SYBR Green/ROX qPCR Master Mix (2X) (ThermoFisher Scientific, Waltham, MA, USA). For each gene, respective forward and reverse primers listed in Appendix A were used according to the manufacturer’s instructions, and *UBQ10* was the reference gene for qPCR. Three biological replicates and three technical replicates per biological replicate were used for each reaction at 12, 32 and 67 dpi. Relative expressions were determined using the 2^−ΔΔT^ method and a two-tailed *t*-test was performed. Results were considered significant if the *p*-value was less than 0.5.

## 3. Results

### 3.1. Comparison between Differentially Expressed (DE) MeWRKYs in Cassava TME3 and T200 at 12, 32 and 67 dpi

Expression of DE WRKYs was validated by RT-qPCR (Appendix A). *MeWRKY3* and *MeWRKY11* were both upregulated at 32 dpi in TME3. In T200, *MeWRKY70* expression was downregulated in T200 at 32 dpi, while *MeWRKY18/59* was upregulated at 67 dpi. These results correlated with the DGE (RNAseq) data. *MeWRKY18/59* expression was not detected by DGE in TME3 at 67 dpi (Table 1) but was downregulated in the qPCR results. While *MeWRKY70* expression was not detected by DGE in TME3, qPCR amplified expression in both healthy and infected plants, but there was no significant difference between the two landraces. *MeWRKY11* was not detected in T200 at 32 dpi by DGE but exhibited low expression in mock-inoculated and infected T200. The reason for discrepancies may be because DGE analysis is less accurate for lowly expressed genes.

Early infection (12 days post infection; dpi) in cassava is the pre-symptomatic stage where virus replication was detected at very low levels or not at all. At 32 dpi, virus titers were high in both landraces, both of which displayed disease symptoms, although symptoms were lower in SACMV-tolerant TME3. In TME3, at late infection (67 dpi), recovery of symptoms was observed (Figure 1A); while in susceptible T200, symptoms and virus load increased (Figure 1B). In total, differential expression of thirteen *MeWRKY* TFs was observed in both SACMV-infected susceptible T200 and tolerant TME3 landraces (Table 1). *Arabidopsis AtWRKY* homologs and their functions are also presented in Table 1. Of the 13 differentially expressed (DE), *MeWRKY*s, four and nine were found in TME3 and T200, respectively, while only *MeWRKY3* was common between the two landraces (Figure 2). Expression of *MeWRKY3* was increased only at 32 dpi in TME3, while in T200 upregulation of *MeWRKY3* relative to a mock-inoculated control occurred at all three stages of infection (Figure 3). No up- or downregulated WRKYs were detected in TME3 at 12 dpi, while in T200, four *WRKY*s were differentially expressed (Table 1; Figure 4). Of the 12 DE *MeWRKY*s in T200 at 32 and 67 dpi, only *MeWRKY44* was downregulated at early infection (12 dpi) and the other eleven were up-regulated. In TME3, three *WRKY*s were up-regulated at 32 dpi and one *MeWRKY81* was uniquely downregulated only in TME3 at recovery (67 dpi) in response to SACMV infection. Furthermore, *MeWRKY11* was only up-regulated (log2 fold change = 2.17) in TME3 at 32 dpi but not in T200. In T200, expression of eight unique *MeWRKY* TFs was uniquely altered in T200 but not in TME3 (Table 1; Figure 4). Notably, *MeWRKY*27 and *MeWRKY55* were up-regulated at all stages of infection in susceptible T200.

### 3.2. Phylogenetic Analysis of Differentially Expressed WRKYs in Arabidopsis, and Cassava T200 and TME3

A phylogenetic tree of AtWRKY transcription factors from *Arabidopsis thaliana* and cassava [*Manihot esculenta* (Me)] based on the phylogenetic tree of Wei et al., 2016 [40] was sorted into taxonomic groups based on amino acid sequences (Figure 5). With regards to phylogeny, most of the AtWRKY homologs described by the Phytozome Plant Comparative Genomics platform were shown to lie topologically close to their cassava counterparts. The highest number (six) of MeWRKYs was represented in Group III. No differentially expressed MeWRKYs were represented in Group IIe and IIb from either T200 or TME3. Group 1c and Group1N had two and one represented DE MeWRKY from T200, respectively, while Group IId had one representative each from T200 and TME3. Interestingly, only MeWRKY11 (AtWRKY40 homolog) from TME3 fell into Group IIa.

### 3.3. Predicted Interactions between AtWRKY Protein Homologs of Differentially Expressed Cassava MeWRKYs and Their Interacting Partners within a Central AtWRKY 33, 40, 53 and 70 Protein–Protein Network

Since there is no experimental *Manihot esculenta* interactome data, we inferred the networks of WRKY-interacting proteins using homologs/orthologs in *Arabidopsis thaliana*. Interactions between AtWRKY protein homologs were derived from differentially expressed cassava MeWRKYs. Interacting partners with the *Arabidopsis* protein homologs within a central AtWRKY 33, 40, 53 and 70 protein-protein network were identified in T200 and TME3 at 12, 32 and 67 dpi. Proteins of differentially expressed gene partners and their functions are represented in Figure 6, Appendix A. In T200 at 12 dpi, *MeWRKY27* and *MeWRKY55* were up-regulated, and AtWRKY33 and AtWRKY53 protein homologs of MeWRKY27 and MeWRKY55, respectively, were found to interact with upregulated HSPRO2 (a R gene ortholog of HS1^Pro−1,2^ in sugar beet) in the central network (Appendix A). In TME3, all interacting protein partners JASMONATE ZIM-domain (JAZ1) transcriptional repressor, cysteine-2/histidine-2-type zinc finger transcription factor (STZ), DICARBOXYLATE CARRIER 2 (DIC2), HSPRO2 and a MAPK phosphatase (AT2G30020) were from significantly downregulated genes (Appendix A). In contrast at 32 dpi, twelve interacting proteins, namely a Pheromone receptor-like-encoding gene (ARF781), Chromatin assembly factor Ib (CAF1b), JAZ1, STZ, tetraspanin8 (TET8), MAPK phosphatase, HSPPRO2, DIC2, ethylene responsive factors ERF5 and ERF6, Mitogen-activated protein kinase 3 (MPK3) and a zinc finger (CCCH-type) transcription factor (CZF1), were from up-regulated genes in T200 (Figure 6A). In TME3 at 32 dpi, *MeWRKY11* and *MeWRKY18* were up-regulated, but only three interacting protein partners (ERF1, JAZ1 and HSPRO2) with the AtWRKY40 and 70 homologs in the network were from upregulated genes (Figure 6B). No downregulation, as was the case with T200, of any interacting partners was observed. At 67 dpi in T200 (Appendix A), several interacting protein partners were similarly from up-regulated genes at 32 dpi, namely STZ, TET8, HSPRO2, DIC2, ERF5, ERF6, MPK3 and CZF1. However, *JAZ1* and a *MAPK phosphatase* (AP2C1; AT2G30020) were not differentially expressed (DE) at 67 dpi, while heat shock factor (*HSFA4A*) was uniquely up-regulated at this time point but not at 32 dpi. *JAZ1* and *ARF781*, which were up-regulated at 32 dpi, were not DE at 67 dpi. Most notably, in TME3 at 67 dpi (Appendix A), none of the *MeWRKY* TFs associated with the AtWRKY33, 53, 40 and 70 homologs in the central network or interacting partners were differentially expressed. Only *MeWRKY81* was downregulated at this recovery stage in TME3 (Table 1).

#### 3.3.1. Functional (GO Term) Enrichment Analysis of Biological Processes Associated with Differentially Expressed Genes in T200 and TME3

Due to the incomplete annotation of genes and their functions in the cassava genome, gene functions were extrapolated from homologs/orthologs identified in the *Arabidopsis* genome. We performed functional enrichment analyses of GOs (Biological Process) on the gene expression data from T200 and TME3 [60]. The overall distribution of adjusted *p*-values was considered, and it was found that the threshold 1 × 10^−8^ for the heat map contained ~20% of the total GO terms. Furthermore, of those 20%, ~85% of enriched GO terms were found between 1.0 and 1.01 × 10^−8^. If the threshold was increased to 5.9 × 10^−11^, approximately 3% of the GO terms were represented and this was deemed too stringent. If the threshold was lowered there were too many GO terms to process. Therefore, 1 × 10^−8^ was considered to be the appropriate cut-off for this study. A heat map of enriched GO terms (adjusted *p* value < 1 × 10^−8^) from differentially expressed (up-regulated/downregulated) gene datasets in susceptible T200 and tolerant TME3 at 12, 32 and 67 days post SACMV infection (Appendix A) was constructed (Figure 7). The ratio of genes within each associated GO term relative to the total set of differentially expressed genes (up or down) is reflected in Figure 7. Fifty one GO terms (biological processes) were found to be enriched across T200 and TME3; however, notably the enriched GO terms were represented only in the upregulated gene sets in both T200 and TME3. No enrichment of biological processes was evident at early infection (12 dpi) in T200, and only four biological processes (BP) (response to chitin, organonitrogen compounds, wounding and cell death), were down regulated in TME3. Notably, there was no overrepresentation of biological processes at the later stage (67 dpi) in TME3, where symptom recovery takes place.

In T200 and TME3 at 32 dpi, overrepresentation of 39 and 28 GO terms were observed, respectively. Forty three biological processes were enriched in T200 at 67 dpi. GO terms associated with response to (JA, GO:0009753), cellular response to JA stimulus (GO:0051716) and JA mediated signaling pathway (GO:0009867) were found to be overrepresented in up-regulated genes in both TME3 and T200 at 32 dpi. However, compared to T200, in TME3 the ratio/proportion of genes associated with these three biological processes was higher compared to T200. Of particular note, the GO term associated with response to JA (GO:0009753) had the highest ratio (0.25) of expressed genes in TME3 at 32 dpi compared to all the gene ratios associated with the other GO terms in both landraces. Seven enriched biological processes were found to be overrepresented within genes differentially expressed in TME3 at 32 dpi, in particular GO terms associated with the ABA signaling pathway (GO:0009788), response to ABA stimulus (GO:0050896), JA biosynthesis and metabolic processes (GO:0006694) and response to ethylene (GO:0009723), were not present in T200. In contrast in T200, seventeen and twenty-two GO terms were overrepresented at 32 and 67 dpi, respectively, that were not present in TME3. Twenty percent of up-regulated genes in T200 at 32 dpi were found to be associated with the biological process, in response to chitin (GO:0010200). In T200 at 32 and 67 dpi, three enriched GO terms were uniquely associated with salicylic acid (SA), namely response to SA (GO:0009751), SA-mediated signaling pathway (GO:0009862) and cellular response to SA stimulus (GO:0071446). These GO terms were not represented in TME3. Other GO terms of interest found to be overrepresented in upregulated genes in T200 at 32 and 67 dpi, but not observed in TME3, were MAP kinase cascade (GO:0000165), signal transduction by protein phosphorylation (GO:0007165), ethylene metabolic and biosynthetic processes (GO:0009692 and GO:0009693) and the respiratory burst (GO:0045730).

#### 3.3.2. Hormone Gene Responses in T200 and TME3

Since differences in overrepresentation in GO terms associated with hormone signaling pathways were observed, the number of genes differentially expressed in T200 and TME3 at each time point that were associated with enriched phytohormone GO terms were calculated. Raw counts of genes that had the hormone represented in their associated GO term (adj. *p*-value < 0.05) were calculated by counting all unique genes that related to a specific hormone derived from multiple GO terms (Figure 8). Interestingly, in tolerant TME3 at 12 dpi, genes associated with ABA, ET, JA, and SA functions/pathways were found to be overrepresented (15–22 counts) amongst downregulated genes (Figure 8). No hormone-related genes were observed in up-regulated genes at 12 dpi. There was also no representation of hormone associated genes (up or downregulated) in T200, at this time point. At 32 dpi, in both T200 and TME3, a number (range of 15–35 counts) of up-regulated genes associated with SA, ABA, JA and ET was observed, with JA and ET exhibiting the highest counts. At 67 dpi, there was no overrepresentation of genes associated with these phytohormones in TME3, but all four pathways had a number (range of 11–27) of up-regulated genes (11–27) in T200, with ABA having the lowest (11) count.

#### 3.3.3. Differentially Expressed Interacting Partners Associated with Upregulated Me WRKYs in Enriched Hormone Pathways

Due to high homology between cassava and *Arabidopsis* WRKYs (Figure 5; Reference [40]), differentially expressed (DE) interacting gene partners associated with *Arabidopsis* homologs (AtWRKY33 and 53 in T200 and AtWRKY40 and 70 in TME3) of upregulated *MeWRKY* TFs in enriched SA, ET, JA or ABA hormone pathways post-SACMV infection could be identified (Appendix A). No hormone enrichment or associated DE genes was observed for TME3 at 12 dpi, while T200 displayed two up-regulated genes, *ATL2* (a putative ubiquitin RING-H2 type ligase) and *AT1G23710* (function unknown but induced after exposure to chitin and early signaling pathway) (Appendix AA). At 32 dpi, enriched hormone functions in T200 were connected to AtWRKY33 and 53 homologs, in particular AtWRKY33, which formed a central hub in T200 (Appendix AB). Enriched hormone pathways included ABA-activated signaling pathway; response to JA; JA biosynthesis and signaling pathways; ET-activated signaling and SA mediated pathways; and response to SA. All interacting partners in the network were up-regulated (Appendix A). Highly interconnected nodes were associated with MPK3, STZ, CAF1b and SYP121 in T200 but were not present in TME3. In contrast, in TME3 at 32 dpi, hormone enrichment was connected to AtWRKY40 and 70 hubs (Appendix AC). JAZ1 and ERF1 were uniquely up-regulated in the network (Appendix AD), which was not observed in T200. Upregulation of JA, SA, ET and ABA related functions were also observed in TME3 but formed part of the AtWRKY40 network not AtWRKY33 as was the case with T200. The number of DE hormone-associated genes in TME3 was significantly lower than that observed for T200. At 67 dpi in T200, the hormone-enriched pathways were similar to 32 dpi, with AtWRKY33 forming the main hub (Appendix AC). Since no DE hormone genes were detected in TME3 (Figure 8) no enriched pathways could be predicted at 67 dpi.

## 4. Discussion

### 4.1. Key Findings

#### 4.1.1. Patterns of WRKY Responses to SACMV Differ in Susceptible T200 and Tolerant TME3 and Favor Upregulation

WRKY TFs have been reported to play important roles in cellular and physiological processes, including pathogen immunity, plant growth/development, senescence and diverse responses to abiotic and biotic stress caused by insect herbivores and pathogens [69,70,71,72,73,74,75]. Herein, we identified for the first time *MeWRKY*s involved in response to cassava mosaic disease caused by SACMV. Similarities and differences between SACMV-responsive *MeWRKY* expression profiles between T200 and TME3 were observed. Susceptible T200 exhibited a greater number (nine) of differentially expressed (DE) *MeWRKY*s compared to only four in TME3 (Table 1) across all three infection stages. Of all the DE *MeWRKY*s at the three stages of infection in both landraces only three were downregulated, demonstrating that overall responses to cassava mosaic disease in both T200 and TME3 landraces appears to favor upregulation of WRKY expression. A general lower number of *WRKY* expression changes in response to SACMV in TME3 compared to T200 may be one hallmark of general metabolic stability and viral stress tolerance. WRKYs can be positive or negative regulators and positively or negatively regulate various aspects of abiotic and biotic stress, PTI or ETI depending on host and elicitor [11]. T200 at 12 dpi exhibited downregulated *MeWRKY44* and up-regulated *MeWRKY*3, 27 and *55* that were not DE in TME3 at this or any other stage of infection; however, no GOs were enriched in T200 (Figure 7). In TME3 no SACMV-responsive DE WRKYs were observed at 12 dpi; however, while response to wounding and cell death GOs were uniquely enriched in TME3 (Figure 7), no link to WRKY expression was established. Concomitantly, these results indicate that WRKY expression is virus-responsive at an early stage in T200 but does not play a direct role in defense. These WRKY TFs may be general stress responders. While the functions of the SACMV-responsive *Arabidopsis* WRKY TF homologs in other plant hosts are involved in resistance towards several bacterial pathogens (Table 1), there was no evidence of WRKY-associated CMD resistance in this study. Evidence pointed to a role of MeWRKYs in hormone regulation and signaling in T200 and TME3 associated with susceptibility and biotic stress tolerance, respectively. Further investigation of the functions of cassava WRKYs is warranted.

#### 4.1.2. MeWRKY TFs Are Associated with Biotic Stress Responses

The most significant finding in both SACMV-infected T200 and TME3 landraces was that WRKY expression changes were associated with biotic stress responses. WRKYs are involved in diverse responses to abiotic and biotic stress caused by insect herbivores and pathogens [69,70]. Despite several studies on plant WRKY responses to insects, and fungal and bacterial pathogens in experimental and annual plant hosts, fewer reports exist on WRKYs in plant virus infection in orphan crops. Expression of *MeWRKY* genes in cassava leaf tissue has been estimated to be 95% (68/72) with about 56% showing high expression levels in leaves [40]. Since many MeWRKYs were shown to have orthologous functions to those in *Arabidopsis* [40], proposed functions of cassava MeWRKYs could be derived in this study. Plants tend to strike a balance between their pathogen response and biotic stress to combat the adverse effects on their growth. In TME3 it is proposed that *MeWRKY* expression responses led to stress tolerance to SACMV while in T200 these stresses were associated with more severe symptoms and higher virus replication. Wounding, regulation of cellular response to stress and regulation of programmed cell death were common enriched stress-related biological responses (Appendix A; Figure 7) to SACMV in both susceptible and tolerant landraces at 32 dpi when virus replication and symptoms increase in both susceptible and tolerant landraces. WRKYs, such as *Nicotiana attenuata* NaWRKY3 and NaWRKY6, have been shown to be involved in response to wounding [72]. In this study, upregulation of *MeWRKY27/68* (*AtWRKY33* homolog) in T200 at 12, 32 and 67 dpi was observed, while it was absent in TME3. Several *Arabidopsis* WRKYs, such as AtWRKY40, AtWRKY33 and AtWRKY70, are known stress regulators. In *Arabidopsis*, upregulated transcription of *MPK3*, *AtWRKY33*, *AtWRKY40* and *AtWRKY70* in response to H_2_O_2_ or other oxidative stresses was observed [73]. Taken together, results herein support a strong regulatory role for MeWRKY27 and 68 in general cellular stress responses to SACMV in T200. AtWRKY33 is a negative regulator of oxidative stress and ABA [72], and up-regulation of the cassava MeWRKY27/68 homologs at 32 and 67 dpi may account for the under-representation of the stress hormone ABA-associated GO terms in T200 (Figure 7). *MeWRKY55* was upregulated across all three time points in T200. The *AtWRKY53* homolog of *MeWRKY55* is a calcium/calmodulin responsive TF implicated in stress signaling in plants as well as being a positive regulator of senescence [73]. These results collectively support a role for these WRKYs in T200 in stress responses to SACMV, which contribute to disease symptoms. In contrast, TME3 only demonstrated downregulation of a single *WRKY* (*MeWRKY81*) at 67 dpi, supporting the proposal that stress responses are reduced at recovery, leading to a reduction in symptoms (Figure 1) and virus load [60]. A number of oxidative stress-related processes, such as cell death, respiratory burst and senescence were observed in T200 at 32 and 67 dpi. *MeWRKY55*, a homolog of *AtWRKY53*, and *MeWRKY27/68* (*AtWRKY33* homolog) were overexpressed at all three stages of SACMV infection in T200. Furthermore, *MeWRKY18/59/83* (*AtWRKY70* homologs) was upregulated in TME3 and T200 at 32 and 67 dpi, respectively. AtWRKY53 and 70 are positive and negative regulators of senescence/stress [52], and AtWRKY33 is positively upregulated by oxidative stress, in *Arabidopsis*. Concomitantly, these results suggest that these WRKYs are involved in oxidative stress responses which contribute to severe CMD symptoms observed at 32 dpi [60]. Further evidence is provided by enriched biological processes and hormone profiles from both cassava landraces that reflect oxidative and other biotic stress responses to SACMV, rather than specific immunity. In TME3 symptom recovery and reduction in virus load is hallmarked by the absence of significant (adj. *p* value < 1 × 10^−8^) enriched GO terms associated with hormone and stress related biological processes (Figure 7). No upregulation of hormone genes were observed in TME3 at 67 dpi (Figure 8). The unique expression changes in *MeWRKY18* (*ATWRKY70*) and *MeWRKY81* (*AtWRKY28*) in TME3 at 32 and 67 dpi, respectively, suggest their associated biological processes (JA signaling and transcriptional regulator of ABA, respectively) may contribute to recovery and tolerance. JA and ABA are well-known stress responsive hormones. While other mechanisms such RNA silencing [76] and ETI-associated resistance (R) genes [77] have been shown to contribute to SACMV tolerance, none of these were directly associated with the DE *WRKYs* identified in T200 or TME3. A study of mes-miRNA expression in TME3 at 67 dpi revealed highly significant downregulation of mes-miR167 [76]. MiR167 targets an auxin responsive factor which plays a role in auxin signaling and adaptive responses to stress. These results support the proposal that symptom recovery in TME3 may be associated with stress tolerance and maintenance of cellular homeostasis. One of the other remarkable features of TME3 was that there were no *WRKY* responses to SACMV at the early pre-symptomatic (12 dpi) stage in contrast to T200, where three (*MeWRKY3, 27, 53*) were upregulated and one (*MeWRKY44*) downregulated. Fewer induced early changes in regulation of WRKY-associated targets in TME3 compared to T200 suggest that an attenuated response to SACMV is a possible hallmark of general tolerance.

Expression of nine *MeWRKY* genes was demonstrated in a study in cassava in response to hydrogen peroxide (H_2_O_2_) [40]. *MeWRKY49, 83* and *89* were downregulated while others were upregulated depending on time after treatment [40]. H_2_O_2_ in plants plays a key role in abiotic and biotic signaling but also its presence in higher levels may promote oxidative damage of biomolecules [78]. There is a complex network of an anti-oxidative system (AOS) for the survival of stressed plants [79], and a characteristic of varied plant stressors is their potential to promote the generation of reactive oxygen species (ROS) in plant tissue which causes oxidative stress. Cells have enzymes which prevent the formation of intracellular H_2_O_2_, for example, glutathione peroxidase and ascorbate peroxidase. *MeWRKY83* (*AtWRKY70*) and *MeWRKY81* (*AtWRKY28*) were upregulated and downregulated at 67 dpi in T200 and TME3, respectively, strongly suggesting a possible role in oxidative stress responses to SACMV infection. Depending on the degree of oxidation, triggered programmed cell death or acclimatization of the plant/increased stress tolerance may occur. Upregulation of several stress related biological processes (Figure 7; Appendix A), which represented a cellular and oxidative stress state, were noted in T200 and TME3 at 32 dpi as disease symptoms and virus replication increased from 12 dpi in both landraces.

Excessive oxidation and reduction of cell components are detrimental and can impair plant growth and development, so maintaining redox homeostasis is crucial [80], and for this reason, plants are rich in compounds with anti-oxidative activity. Evidence for oxidative stress tolerance in cassava in response to SACMV is a key finding. The most prominent antioxidants in plants are water-soluble ascorbate (Asc), glutathione (GSH), phenols and liposoluble tocopherols. Peroxidases, for example glutathione peroxidases (GPXs), are important in ROS detoxification since they catalyze the reduction of H_2_O_2_ and protect cells and enzymes from oxidative damage. Plant glutathione-S-transferases (GSTs) catalyze the conjugation of glutathione (GSH) to a wide variety of compounds and also perform a variety of catalytic and non-enzymatic functions in normal plant development [81]. GSTs form a network with several GPXs and both play important roles not only in various biological processes including cellular growth and development but also expression of stress responsive genes [82]. In cassava, GST (tau8) forms an interactive network with GPXs 1–8, and its expression was found to be upregulated in T200 and TME3 [60]. GST was also found to respond to a ssRNA virus (*Cassava brown streak virus*) in both susceptible (Albert) and resistant (Kaleso) cassava varieties [83]. This supports the notion that oxidative damage and stress hormone responses are common responses to virus infections in cassava. Ethylene-related biological processes were significantly enriched in both T200 and TME3 at 32 dpi and T200 at 67 dpi. GPXs are also involved in abiotic and biotic stress responses as a result of combined salicylic acid and ethylene signaling [84], providing further support for the notion that stress responses to SACMV in T200 and TME3 are an attempt to limit oxidative damage.

Search of potential candidate genes in the major quantitative trait locus (QTL) region associated with dominant homozygous CMD2 resistance in cassava identified two peroxidases and a thioredoxin gene [85]. Plant thioredoxins and peroxidases play a fundamental role in tolerance of oxidative stress [86]. This lends credence to the results from this study that responses of cassava to CMD are associated with biotic stress, and that responses to oxidative stress in TME3 are key to tolerance to *South African cassava mosaic virus*. For a perennial crop such as cassava, and maybe others, this may be a strategy to mitigate the cost of a reduction in long term growth as a consequence of mounting an energy-costing resistance response against viral pathogens. While no WRKY TFs have been previously identified in association with CMD, a major novel QTL for bacterial blight (caused by *Xanthomonas axonopodis* pv. manihotis) resistance was reported in cassava. DNA sequence analysis of the QTL intervals revealed 29 candidate defense-related genes, and two of them contain domains related to a R gene (NB-ARC-LRR) and WRKY TF [87]. A further study showed that MeWRKY75 and Whirly transcription factors (MeWHY1, 2 and 3) interact and confer resistance to bacterial blight in cassava [42]. Further in vivo studies and gene editing to identify WRKYs that could be manipulated to provide resistance to cassava viruses is warranted.

#### 4.1.3. SACMV-Responsive WRKYs Are Associated with Phytohormones and Enriched Hormone Biological Processes (GOs)

The most compelling evidence for SACMV-responsive WRKYs role in the etiology of CMD comes from the enriched hormone GOs and hormone results (Figure 7 and Figure 8). From these results, we conclude that the WRKYs in TME3 and T200 are regulators of stress responses that lead to JA, SA, ABA and ET responses, which are classic biotic stress markers. Biotic and abiotic stress in plants activates SA, JA, and ET and ABA hormone pathways which subsequently change the transcription level of related genes and translated proteins [72,74]. WRKY transcription factors play an important role in cross-talk between regulatory mechanisms of plant pathogen defense and general biotic stress responses. Regulation of WRKY TFs and their actions involves phytohormones, other transcription factors and proteins that interact through complex networks. While JA, SA, ABA and ET are also known to be involved in defense responses to pathogens, results in this study provide no evidence of classic defense hallmarks. We conclude that the WRKY and hormone responses are associated with activation of general oxidative and other biotic stress responses in both landraces at 32 dpi. In T200 this does not lead to SACMV tolerance, whereas in TME3 disease symptoms are partially mitigated (tolerance). Since the majority of WRKYs in T200 and TME3 were upregulated, we propose that these TFs are mainly positive regulators of stress responses. Cassava does not demonstrate a hypersensitive response (HR), typical of defense associated with many other plant virus-induced SA responses and systemic acquired resistance (SAR). Unravelling virus tolerance mechanisms in this non-model orphan crop may be a requisite for adopting a different approach needed for genetic engineering virus resistance.

At 12 dpi there were no highly significant enriched GO terms in T200 (Appendix A) due to a very stringent cut off (adj. *p* value < 1 × 10^−8^) chosen for our heat map (Figure 7). However, at adj. *p* value < 0.02 (Appendix A) there were 14 represented GOs in T200, with a number of upregulated genes that were associated with stress such as wounding, stress and the respiratory burst and a number of downregulated genes associated with growth and meristem regulation (Appendix A). In contrast, in TME3 four GOs (response to chitin, wounding and organonitrogen compounds and cell death) were highly enriched. We conclude that in the early pre-symptomatic stage both landraces respond to SACMV by stress-related processes, but in TME3 these may be more effective in reducing the impact of SACMV infection. There were no DE WRKYs detected at 12 dpi in TME3, but interestingly a number of hormone-associated genes (ABA, SA, JA and ET) were downregulated but not in T200 (Figure 8). These results indicate that this early hormonal response in TME3, while not associated with WRKY responses, may be key to SACMV tolerance in TME3. In contrast, in T200, four responsive WRKYs were detected (Table 1), but there were no differentially expressed (DE) hormone genes detected. We propose that in T200 the gene targets of these early DE WRKYs contribute to higher susceptibility as T200 develops higher viral loads and symptoms compared to TME3 (Figure 1) at 32 dpi and thereafter.

At 32 dpi, there were a number of ABA and SA genes upregulated in both T200 and TME3, but the number of upregulated ET and JA acid genes were higher in both susceptible and tolerant landraces (Figure 8). ET, JA, ABA and SA all respond to biotic and oxidative stress and results herein provide evidence that cassava responds to SACMV replication by biotic stress-induced upregulation of hormones. Notably, WRKY expression is mostly upregulated in T200 and TME3 (7 out of 8) at this time point and we conclude that these MeWRKYs are positive regulators of biotic stress. These conclusions are supported by the observation of several stress-related GO terms, such as response to wounding, cell death, cellular response to stress and chitin and ‘plant-type’ hypersensitive responses in both landraces (Figure 7; Appendix A).

Interestingly, several enriched immune-related biological processes were observed in T200 and TME3 at 32 dpi. Biotic stress and pathogen plant responses are known to overlap significantly [72,74], but in cassava evidence points to more prevalent roles for WRKY and hormones in stress responses. AtWRKY53 and AtWRKY70 are both positive and negative regulators, respectively, of senescence in *Arabidopsis* [51]. The cassava homologs, *MeWRKY15* and *MeWRKY55*, of *AtWRKY53* were upregulated at 32 dpi in T200, while *AtWRKY70* homologs *MeWRKY59* and *MeWRKY83* were upregulated at 67 dpi. Furthermore, *AtWRKY53* and *AtWRKY70* are upregulated in response to oxidative stress (H_2_0_2_) [88], and cassava homolog *MeWRKY55* of *AtWRKY53* was upregulated at all three stages of infection in T200, while at recovery in TME3 *MeWRKY18* (*AtWRKY70*) was upregulated. From these results, we conclude that cassava homologs of *AtWRKY53* and *AtWRKY70* respond to oxidative stress. We propose that *MeWRKY15/55* and *MeWRKY59*/*83* are positive regulators of senescence or cell death related biological processes in T200 at 32 and 67 dpi, respectively. Expression in wild-type and SA-deficient *Arabidopsis* mutants suggests a role for SA in induction of *AtWRKY53* and *AtWRKY70* in senescence, and the observation of enriched SA GOs (Figure 7) uniquely in T200 at 32 and 67 dpi corroborates a role for these WRKYs in susceptibility.

The phytohormone ABA is the best-known stress signaling molecule in plants and also protects plants from biotic and abiotic stresses. Notably, while the number of ABA genes were upregulated in T200 and TME3 at 32 dpi, ABA processes such as ABA-activated signaling pathway and cellular response to ABA stimulus, were only significantly represented in tolerant TME3 (Figure 7). The interaction between WRKYs and ABA in stress is largely unknown. ABA can enhance plant antiviral defense as shown for several viruses. Interplay has been reported between ABA signaling and RNA silencing, which interferes with virus accumulation leading to plant viral resistance [89]. WRKY8 in *Arabidopsis* was shown to function in defense of TMV-cg in *Arabidopsis* by mediating both ABA and ET signaling. Exogenous applications of ABA also reduced systemic accumulation of TMV-cg [53]. While in *Arabidopsis* AtWRKY40 (MeWRKY11) acts as a negative regulator and binds to multiple ABA-inducible genes leading to inhibition of expression, in TME3 at 32 dpi, MeWRKY11 was uniquely upregulated and not in T200, suggesting it is a positive regulator. AtWRKY40 can also interact with AtWRKY18 and AtWRKY60 in *Arabidopsis* to inhibit the expression of crucial stress-responsive genes. We propose that at around 32 dpi, TME3 begins to show some signs of recovery as symptoms are generally lower compared to T200. Cassava is a perennial crop and ABA induction may be important for long term tolerance to virus infection.

In the case of T200, overrepresentation of SA-related biological processes and absence of significant ABA enriched GOs was observed. The effects of ABA are multifaceted and depend on the pathosystem and the timing of induction [90]. Studies have shown that hormone signaling mediated by ABA can promote abiotic stress tolerance and suppress signaling of the biotic stress-related hormone SA [91], strongly suggesting that in T200 SA signaling is repressed by ABA leading to susceptibility. Interestingly, recent studies have also linked ABA to the microRNA (miRNA) pathway through which ABA affects the maturation and stability of miRNAs [92]. ABA was shown to induce resistance against *Bamboo mosaic virus* through Argonaute 2 (AGO2) and AGO3 [93]. In cassava, AGO2 was shown recently to be involved in susceptibility in T200 and an early tolerance response in TME3 [76]. The accumulation of a number of small RNAs in plants is affected by ABA and abiotic stresses [94]. The miR168-mediated feedback regulatory loop regulates ARGONAUTE1 (AGO1) homeostasis, which is crucial for gene expression modulation and plant development. MIR168 controls AGO1 homeostasis during ABA treatment and abiotic stress responses in *Arabidopsis*. Notably, at early infection, downregulation of cassava mes-miR162 and mes-miR168 that target antiviral posttranscriptional gene silencing (PTGS) regulators *DCL1* and *AGO1*, respectively, was observed in TME3 at 12 dpi, and *AGO1* and *DCL1* expression was higher compared to T200 post infection [76], providing further evidence that ABA and RNA silencing are important in SACMV tolerance in cassava.

Several SA-associated biological processes (BPs) were enriched in both cassava landraces, however, they were more significantly represented in T200 at 32 dpi (3 GOs) and 67 dpi (4 GOs). Since T200 is highly susceptible to SACMV, it is proposed that SA and SA-signaling pathways play a greater role in severe susceptibility to SACMV in T200 than TME3. We further conclude that SA induces a biotic stress response rather than SAR. Since SA biological processes continued to be overrepresented and hormones upregulated at 67 dpi (Figure 7 and Figure 8) in T200, they are most likely to be hallmarks of persistent stress to high virus load. They also represent a metabolic and energy cost to the plant resulting in growth perturbations and severe disease symptoms.

Since ET and JA biological processes (BPs) are highly represented in both landraces at 32 dpi but only in T200 at 67 dpi, we suggest that these hormones are also principally associated with biotic stress at 32 dpi as there was no evidence for CMD resistance responses. Jasmonic acid (JA) also has interconnecting roles with ABA and SA in plant stress and pathogen response. Notably, however, the DE WRKYs in TME3 at 32 dpi share only two upregulated TFs (*AtWRKY41* and *70* homologs) with T200, which implies that a different set of WRKYs respond to SACMV in T200 and TME3 even though they share a similar stress response. We conclude that these WRKYs are positive activators of JA, but in TME3 we suggest JA concomitantly plays a greater role in temperance of disease symptoms (Figure 2) and lower virus loads [60]. Evidence is provided by the observation that TME3 exhibited the highest ratio of JA genes expressed at 32 dpi and five significantly enriched JA BPs compared to T200 (three enriched JA GOs) (Figure 7). Furthermore, the ratio/proportion of genes associated with JA biological processes was significantly higher compared to T200. Of particular note, the GO term associated with response to JA (GO:0009753) had the highest ratio of any expressed genes in TME3 compared to all represented GO terms in both landraces. These results suggest that JA may play a role in metabolic stabilization during infection at 32 dpi in TME3 which contributes to recovery and tolerance later at 67 dpi. AtWRKY28 is a positive regulator of JA signaling [43]. The unique downregulation of *MeWRKY81* (*AtWRKY28*) at 67 dpi is accompanied by a lack of significant JA associated genes in TME3 (Figure 8), and its proposed role in long term tolerance needs further investigation. Over-activation of jasmonate responses can lead to carbon starvation and reduced growth. Collectively, our findings suggest the emergence of diverse strategies in TME3 to keep metabolic disturbances at bay. This study provides new insights into metabolic processes that underlie growth–defense trade-offs in a perennial crop which is critically important for many other orphan crops.

Ethylene is regarded as a stress-responsive hormone besides its roles in regulation of plant growth and development [95] and results clearly demonstrate this hormone is SACMV-responsive. While enrichment of ET-associated signaling pathways and biological processes were overrepresented in T200 and TME3 at 32 dpi, T200 exhibited a higher number (seven) of enriched ET GOs at 32 dpi compared to TME3, and these enrichments persisted at 67 dpi. Upregulation of *MeWRKY27* (*AtWRKY33*) and *MeWRKY15/55* (*AtWRKY53*) uniquely in T200 was associated with other proteins in the AtWRKY33, 53, 40 and 70 interacting network.ERF5 and 6 from this network were upregulated in T200 at 32 and 67 dpi (Figure 6A; Appendix AA) while ERF1 was upregulated in TME3 at 32 dpi. These results, and enriched ethylene processes in both T200 and TME3 at 32 dpi (Figure 7; Appendix A), support a role for ethylene in general biotic stress responses during high virus replication levels at 32 dpi in infected cassava. Furthermore, at 67 dpi three ethylene (ET) GO terms are significantly enriched in T200 but notably not in TME3. Olefin and alkene metabolic and biosynthetic processes associated with ethylene biosynthesis were significantly overrepresented in T200 at 32 and 67 dpi. Since TME3 recovers at around 67 dpi and T200 does not, we conclude that enriched SA and ethylene processes are persistent responses by T200 to survive stress due to high SACMV replication. Severe symptoms are a manifestation of persistent biotic stress responses in T200. None of these enriched SA or ET processes or upregulation of hormone-associated genes (Figure 8) were represented in TME3 at 67 dpi, and we conclude that TME3 exhibits an overall higher metabolic homeostasis or stability and reduction in virus-induced stress. SACMV-tolerant TME3 is able to mitigate energy costs leading to growth recovery and a reduction in symptoms and virus load. This fine balance or tuning of multiple WRKY regulated gene networks and hormone responses warrants further study in this orphan crop.

### 4.2. WRKY Protein-Protein Networks

WRKY transcription factors can regulate diverse plant responses to pathogens and stress through complex interconnecting networks [96]. WRKY proteins play prominent roles in the regulation of transcriptional reprogramming associated with plant stress responses and function via protein-protein interactions and even cross regulation and autoregulation. Collectively, in addition to biological processes, interacting network partners (Figure 6; Appendix A) associated with up or down regulated *MeWRKY* expression demonstrate differences that hallmark susceptibility in T200 and tolerance in TME3. Interactions among WRKY proteins can be with other TFs, other defense proteins or regulatory and signaling molecules. Notably, protein-protein interacting networks demonstrated that interacting AtWRKY33, 53, 40 and 70 partners JAZ1, DIC and STZ were downregulated and these genes may play an early role in tolerance in TME3. In contrast, in T200 only a heat shock protein (HSPRO2) was upregulated in the WRKY network (Appendix AA), and amongst the enriched hormone pathways only a chitin-induced gene (At1G23710) of unknown function and a putative RING-H2 type ligase were represented (Appendix AA). A number of hormones, including JA, were uniquely downregulated in TME3 but not T200 at 12 dpi (Figure 8). JA-Ile activates defense responses by triggering the degradation of JASMONATE ZIM DOMAIN (JAZ) transcriptional repressor proteins, but JAZ repressors of metabolic defense were shown to promote growth in *Arabidopsis* [97]. Plant immune responses mediated by the hormone jasmonoyl-L-isoleucine (JA-Ile) are metabolically costly and often linked to reduced growth and development. We conclude that downregulated *JAZ1* expression and a number (18) of downregulated JA associated genes in TME3 at early infection may mitigate impacts of SACMV on growth, partly by minimizing detrimental metabolic effects.

Ethylene responsive factors (ERFs) were also shown to respond to the geminivirus, tomato yellow leaf curl virus (TYLCV), through a complex network of positive and negative expression [98]. In a T200 protein-protein interaction network with AtWRKY33 and 53, ERF5 and 6 interactors were found to upregulated at 32 and 67 dpi, while in TME3, ERF1 and JAZ1 were interactors in network of upregulated AtWRKY40 and 70. At 32 dpi, ERF5 and 6, and ERF1, were upregulated in T200 and TME3, respectively (Figure 6A,B), proposed to be a result of elevated ethylene biosynthesis pathways (Figure 7). We conclude that these stress related ET-responsive factors trigger downstream biological processes which contribute to disease symptoms and negatively affect plant growth in T200 and TME3. Two other WRKY protein network partners of note were two upregulated transcription factors STZ and CZF1 that may be involved in regulation of a number of biological processes in T200 and TME3 such as the respiratory burst, response to wounding and mechanical stimulus, response to hormones (ET, JA and ABA) amongst others (Appendix A). Downregulation of STZ in TME3 at 12 dpi and upregulation of STZ and CZF1 in T200 at 32 and 67 dpi results in downregulation and upregulation of their target genes in T200 and TME3, respectively. We conclude that these two TFs as part of the network are SACMV-responsive and contribute to the symptom and plant growth phenotypes of the two landraces. Plant growth and leaf development is compromised to a greater degree in susceptible T200 compared to TME3 (Figure 2).

Detailed studies on the mechanisms of signaling and transcriptional regulation have unveiled the association of WRKY proteins with mitogen-activated protein kinases (MAPKs), MAPKKs, 14-3-3 proteins, calmodulin, histone deacetylases, disease-resistance proteins and other WRKY TFs [99]. Of some of the interacting proteins in the DE WRKY central network, AtWRKY25 requires Zat7 and ascorbate peroxidase (APX_1_) to induce the oxidative stress response [100]. Mitogen-activated protein kinase (MAPK3) is also responsive to oxidative and other biotic stress, and notably MPK3 was upregulated only in the protein-protein network in T200 at 32 dpi (Figure 6A) and 67 dpi (Appendix AA) and was also represented in the enriched hormone pathways induced by SACMV at 32 (Appendix AB) and 67 dpi (Appendix AC). Two other partners in the network, SYP121 and CAF1b, were upregulated and are also involved in the MAPK cascade in T200 at 32 and 67 dpi. We conclude that MPK3 is induced either directly by SACMV or indirectly via MPK3-induced oxidative stress and plays a role in severe symptom development, as in contrast MPK3 is not activated in TME3 and symptoms are milder compared to T200. MAPK3 also activates *WRKY* expression in many plants, including *AtWRKY33* [98]. Expression of the *AtWRKY33* homolog, *MeWRKY68*, was upregulated in T200 at 32 and 67 dpi, while the *MeWRKY27* homolog was upregulated at all three stages of infection. We conclude that *MeWRKY27* regulated genes play a role in early infection in T200 which is associated with susceptibility. MAPKs play important roles in transduction of downstream signals in ABA-dependent stress response [101]. The MPK3/MPK6 cascade along with the downstream WRKY TF has been depicted to induce ethylene production through regulation of ACC synthase activity [102]. Collectively these results support the proposal that the central WRKY network and partners play a greater role in establishment of disease symptoms in T200 via stress-induced pathways associated with ethylene and other senescence-related processes (Figure 7; Appendix A). However, further research is required around signaling pathways up and downstream of MPKs, as the knowledge of these signaling cues can impose further improvements in designing stress-tolerant transgenic crops.

### 4.3. Phylogeny and Functions of Cassava WRKY TFs

An unrooted neighbor-joining phylogenetic tree based on multiple alignments of predicted amino acid sequences of the WRKY domains in cassava and *Arabidopsis* [40] was used to map T200 and TME3 SACMV-associated differentially expressed (DE) WRKYs. Overall mapping of proteins encoded by DE *MeWRKY*s in T200 and TME3 (Figure 5) illustrated similar and distinct profiling to WRKY phylogenetic groups, which is proposed to be associated with genetic differences that contribute to SACMV susceptibility or tolerance. Phylogenetic and abiotic experimental analyses [40] revealed there were closely related orthologs between cassava and *Arabidopsis* suggesting an ancestral set of *WRKY* genes existed prior to divergence of these two plants, allowing for identification of possible functions of WRKYs in cassava. Group I and II members contain C2H2 zinc finger motifs, whereas WRKY proteins in group III have C2HC zinc fingers. Different phylogenetic groups had representatives of SACMV-responsive WRKYs, except for Groups IIb and IIe that had no DE WRKYs represented in either landrace. Group II WRKY members contain C2H2 zinc finger motifs, and upregulated *MeWRKY11* (*AtWRKY40* ortholog) in the small subclade in Group IIa WRKYs was uniquely represented in TME3 at 32 dpi. *AtWRKY40* has been shown to play partly redundant functions in regulating plant disease resistance [103]. TME3 also uniquely showed downregulation of *MeWRKY81* (*AtWRKY28* ortholog) at recovery. Neither of these two *WRKY*s were expressed in T200 at any stage of infection. These observations suggest that MeWRKY11 and MeWRKY81 could play positive and negative regulation roles at 32 and 67 dpi, respectively, in priming TME3 for recovery and tolerance post the symptomatic infection stage.

AtWRKY28 (MeWRKY81ortholog) is a regulator of jasmonic acid (JA) signaling [43] and is involved in salicylic acid (SA) signaling [55]. Interestingly, several interacting protein partners in the AtWRKY28 network (Appendix A), such as the DNA binding transcription factors and calmodulin binding proteins CBP60G and SARD1, play redundant critical roles in salicylic signaling [104]. Since SA biological processes were not significantly enriched in TME3 at 67 dpi (Figure 7), it is suggested that downregulation of *MeWRKY81* in TME3 is associated with the absence of a significant SA response. AtWRKY28 is also a transcriptional regulator in *Arabidopsis* of JA and ET signaling pathways, and at 67 dpi there was no significant JA or ET GO enrichment in TME3. The lack of DE hormone genes in TME3 at 67 dpi (Figure 8) suggests metabolic homeostasis. Overexpression of At*WRKY*28 has also been shown to be upregulated under oxidative stress [105,106], and there was no evidence of enriched oxidative stress related GOs (biological processes) (Appendix A) in TME3 at 67 dpi. Collectively, these results provide strong evidence that downregulation of *MeWRKY81* plays a role in reducing oxidative stress at symptom recovery in TME3. This conclusion is also supported by the findings that in contrast to TME3, in susceptible T200 at 67dpi there are several overrepresented (upregulated) GOs for processes such as response to wounding, respiratory burst, regulation of cellular response to stress and cell programmed cell death that were also enriched at the systemic infection stage (32 dpi). Notably, a cassava R gene homolog (Manes05G169600) of the *Arabidopsis* R gene *PRW8-NBS* was upregulated (log2fold 1.42; *p* value 0.05) in TME3 at 32 dpi [60]. AtWRKY28forms part of the PRW8-NBS hub (Appendix A), and it is reported that RPW8 recruits components of basal defense for powdery mildew resistance in *Arabidopsis* [107]. Further studies on the potential role of this cassava PRW8 protein-AWRKY28 network in tolerance to SACMV in TME3 would be both interesting and informative. 

*MeWRKY44* (*AtWRKY7* ortholog) and *MeWRKY*70 (*AtWRKY12*) were the only two downregulated *WRKY*s in T200 at 12 and 32 dpi, respectively, and fell into Group IId and IIc, respectively, but the role of their gene targets in T200 needs to be investigated in future in vivo experiments.

#### Group III MeWRKYs Play a Role in Disease Development in T200 and TME3

The largest numbers (six) of DE *MeWRKY*s in T200 were found in Group III (Table 1; Figure 5). Group III WRKY proteins are evolutionarily dynamic and play a role in plant adaptation and importantly also participate either negatively or positively in basal pathogen defense. Group III *MeWRKY 15, 55, 59* and *83* were upregulated in T200 at 32 or 67 dpi, and we conclude they are negative defense regulators of defense in this landrace. Similarly, infection of tomato with the geminivirus, *Tomato yellow leaf curl virus*, was also associated with six upregulated WRKY Group III TFs that responded to infection [98]. In contrast, *Arabidopsis* WRKY group III transcription factors AtWRKY53 and AtWRKY70 (orthologs of SolyWRKY80/81 and SolyWRKY54/53, respectively), have been shown to be positive regulators of plant defense signaling pathways [108]. Other WRKY TFs that act as negative or positive regulators of defense signaling include AtWRKY7, 40, 41 and 53 [31]. Dual functionality was shown for AtWRKY41, as *Arabidopsis* plants overexpressing *AtWRKY41* showed enhanced resistance toward virulent *Pseudomonas syringae* but decreased resistance toward *Erwinia carotovora* [47]. In T200, *MeWRKY3* (*AtWRKY41*) and *MeRKY55* (*AtWRKY53*) were upregulated at 12, 32 and 67 dpi suggesting they are negative regulators of defense signaling contributing to susceptibility. At early infection (12 dpi) in T200, *MeWRKY44* (*AtWRKY7*) on the other hand was downregulated, suggesting that MeWRKY44 is a positive regulator of defense whose expression is repressed by SACMV. In T200, *MeWRKY3* (*AtWRKY41*) was upregulated throughout the infection stages, while only upregulated at 32 dpi in TME3. T200 and TME3 exhibited severe symptoms and high virus loads at 32 dpi suggesting *MeWRKY3* is a negative regulator of SACMV defense. In T200 at 32 and 67 dpi, and in TME3 at 32 dpi, GO enrichment (upregulation) of responses to salicylic acid, salicylic acid mediated signaling pathway and cellular response to salicylic acid stimulus were associated with Group III upregulated WRKYs. Notably, *MeWRKY59* and *83* (*AtWRKY70* homolog) were upregulated at 32 and 67 dpi in T200, while *MeWRKY18* was upregulated only at 32 dpi in TME3. These three TFs are all homologs of *AtWRKY70* which regulate the balance between JA-dependent and SA-dependent responses [45,46]. JA and SA processes were concomitantly upregulated in T200 at 32 and 67 dpi (Figure 7 and Figure 8; Appendix A) while only in TME3 at 32 dpi. In TME3 at symptom recovery no enriched SA or JA enriched biological processes were observed. These observations provide strong evidence that cassava homologs of AtWRKY70 may show dual functionality, being positive regulators of tolerance in TME3 but negative regulators of tolerance in T200. Overall, results from this study show strong evidence for the role of Group III MeWRKYs in disease symptom development in T200 and TME3.

### 4.4. WRKY Expression Is Virus–Host Interaction Dependent

A comparison of the change in expression of *AtWRKY* homologs of *MeWRKYs* between cassava T200 and TME3, and three other susceptible hosts, potato, *Arabidopsis* and tomato [83,109,110,111,112,113,114,115] infected with four different geminiviruses was undertaken (Appendix A). Results showed that virus–host interactions exhibited different WRKY regulation patterns at different infection stages, providing supporting evidence for a level of WRKY-host-virus co-evolution specificity as well as a temporal regulation. *AtWRKY40, 41, 53* and *70* were amongst the most commonly DE *WRKY*s in the four host plants, and three (*AtWRKY41, 53* and *70*) were found in SACMV-responsive T200. These DE WRKY regulators are commonly observed in many plant host studies and are widely involved in many growth and development processes. Host WRKY responses to pathogens are likely to represent induced metabolic perturbations in many cases that occurred during host-pathogen co-evolution. *AtWRKY11, 23, 27, 30, 65* and *72* were the least responsive to virus infection, and interestingly none were represented in T200 and TME3. Not surprisingly, *AtWRKY70* was the most highly DE in the four virus-infected plants at early, middle and late infection stages of infection as it is involved in multiple functions such as senescence, abiotic and biotic stress, and additionally is a node of convergence for jasmonate-mediated and salicylate-mediated signals in plant defense. Notably, no upregulation of any *WRKY*s were observed in the model host *Arabidopsis* which is highly susceptible to a large number of viruses. *AtWRKY12*, *22* and *28* were downregulated in SACMV-infected susceptible *Arabidopsis*, whereas *AtWRKY12* was only downregulated in T200 at 32 dpi and *AtWRKY28* was downregulated in tolerant TME3 at 67 dpi. *AtWRKY22* was uniquely downregulated by SACMV in *Arabidopsis*. *AtWRKY40* (*MeWRKY11* homolog) is also of interest, which was not differentially expressed in either susceptible cassava T200 or *Arabidopsis* at any stage of infection but uniquely upregulated in TME3 providing further evidence for a role of this TF in tolerance to SACMV infection. Overall, we conclude that virus-WRKY-host interactions have co-evolved over time and in some cases are positive regulators while in others they are negative repressors of target genes that are involved in virus resistance or susceptibility.

*Cassava brown streak virus* and *Ugandan brown streak virus* are ssRNA ipomovirsues, while SACMV is a ssDNA geminivirus. Interestingly, a comparison of DE At*WRKY*s in cassava between these three viruses revealed different upregulated or downregulated WRKY expression results at different infection stages (Appendix A), again demonstrating the specificity between transcription factor, virus and host co-evolution and the growth stage of the plant host. *Manihot esculenta* Crantz originates in South America and was introduced into Africa a few hundred years ago. Since cassava infected with SACMV, CBSV and UCBSV did not share many common *AtWRKY* homolog responses, and in some instances the *WRKY*s showed opposite expression (up or down), we propose that different cassava germplasm is adapting to indigenous viruses from the sub-Saharan African continent.

## 5. Conclusions

To ensure a successful long-term infection cycle, geminiviruses must temper their destructive effects on their host cells and prevent drastic plant responses. In perennial crops such as cassava, this is of utmost importance. Results from this study support the conclusion that, while different WRKY expression profiles were observed in susceptible T200 and tolerant TME3, expression alterations in *WRKY* TFs are associated with stress responses and associated hormone changes. At 32 dpi, both landraces exhibited high virus replication, disease symptoms and overlap in some enriched biological processes, however, *MeWRKY3* was the only common upregulated WRKY in T200 and TME3. No evidence was provided for a direct role of DE WRKYs in defenses such as PTI or ETI. We conclude that in TME3 early infection is linked to repression of hormones ET, SA, ABA and SA which may be associated with reduced SACMV-induced stress, but no link to changes in WRKY expression could be made. Early repression of SA and lack of enriched SA biological processes in TME3 compared to T200 are proposed to be hallmarks of tolerance. Pathway enrichment analysis associated with downregulation of SA signaling and SA-related metabolites is also a strategy of citrus greening disease tolerant citrus cultivars [116]. Tolerance is a preferred option to the energy-cost of long-term active defense. At the early infection stage, T200 susceptibility is associated with several DE WRKYs whose precise roles in early infection are not clear but may be involved in the negative regulation or suppression of hormone and defense responses. Differentially expressed WRKY-associated GO enriched biological processes in T200 clearly demonstrated that they were involved in long term susceptibility as they continued to be enriched at 67 dpi. In contrast, in SACMV-tolerant TME3 at symptom recovery there were no DE WRKY-associated pathway enriched processes, supporting a stasis which represents metabolic stability and CMD tolerance. Collective results from the WRKY expression profiles, and associated protein network partners and concomitant biological process responses to SACMV, proposes a model (explained in detail in the figure legend) for WRKY associated responses to SACMV in T200 and TME3 at 32 dpi (Figure 9A–C). In the absence of any WRKY-associated ETI or PTI responses observed in T200 or TME3, and since all WRKYs (except for *MeWRKY70*) were upregulated in both landraces at the systemic infection stage, we conclude that these TFs play a negative role in downstream defense pathways and a positive role in biotic and oxidative stress responses. These findings highlight the importance of WRKY factors in transcriptionally reprogramming plant responses towards a geminivirus in cassava.

Further in vivo studies on the role of plant WRKYs specifically in response to viruses are required as viral pathogen studies are underrepresented in comparison to bacteria and fungi. Unravelling the regulation of these WRKYs and their network partners will provide further clues to the complex molecular mechanisms of susceptibility and tolerance to CMD. Since WRKY network-virus interactions are variable in different plant hosts at different stages of infection, designing resistance to viruses by targeting a specific WRKY or WRKY hub will be challenging. A large number of WRKYs have been functionally characterized in model plants, providing abundant functional references for other plants. Application of new genetic technologies such as gene editing will accelerate the research progress of WRKY’s novel functions. Reverse genetics experiments are difficult and time consuming for cassava, since the crop is not amenable to transient assays and genetic modification. Recently a CRISPR high throughput platform using cassava protoplasts to knock out genes has been developed [117], and this may prove useful in the future to confirm the roles of MeWRKY transcription factors in disease tolerance or susceptibility.

## Figures and Tables

**Figure 1 viruses-13-01820-f001:**
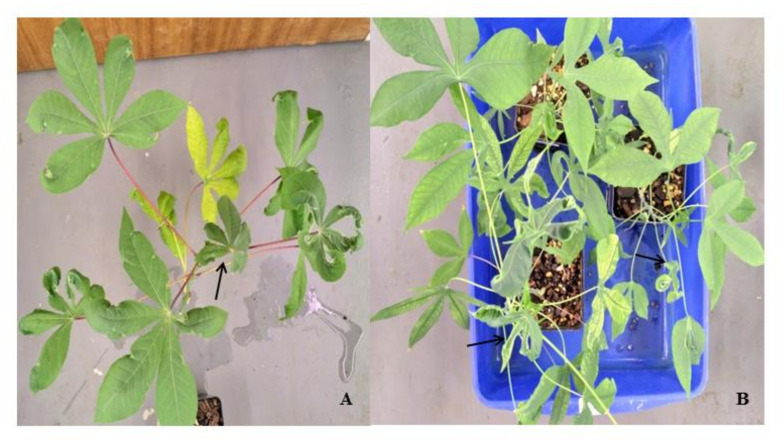
(**A**) Recovery leading to no or milder symptoms in emerging apical leaves (arrow) in CMD-tolerant TME3 (**B**) T200 exhibiting severe symptoms persist (arrows).

**Figure 2 viruses-13-01820-f002:**
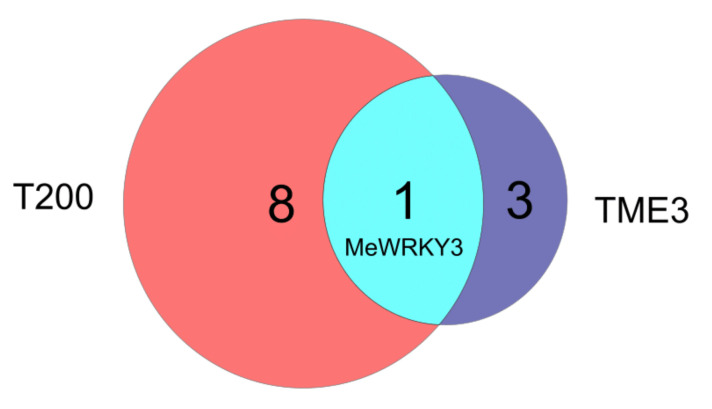
Total number of all *MeWRKY* transcription factors differentially expressed in TME3 and T200 cassava landraces at 12, 32 and 67 days post *South African cassava mosaic virus* infection.

**Figure 3 viruses-13-01820-f003:**
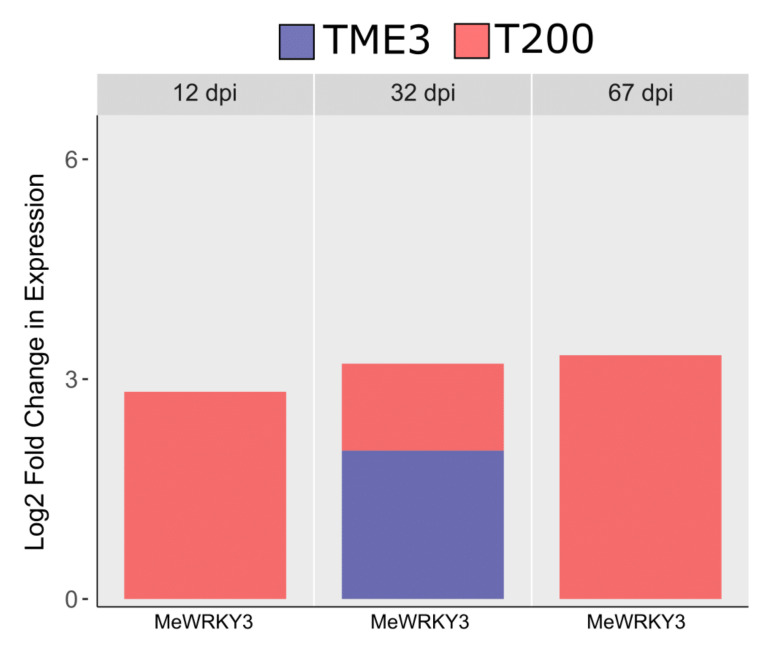
Log2 fold change in expression of *MeWRKY*3 at 12, 32 and 67 days post infection (dpi) with *South African cassava mosaic virus*.

**Figure 4 viruses-13-01820-f004:**
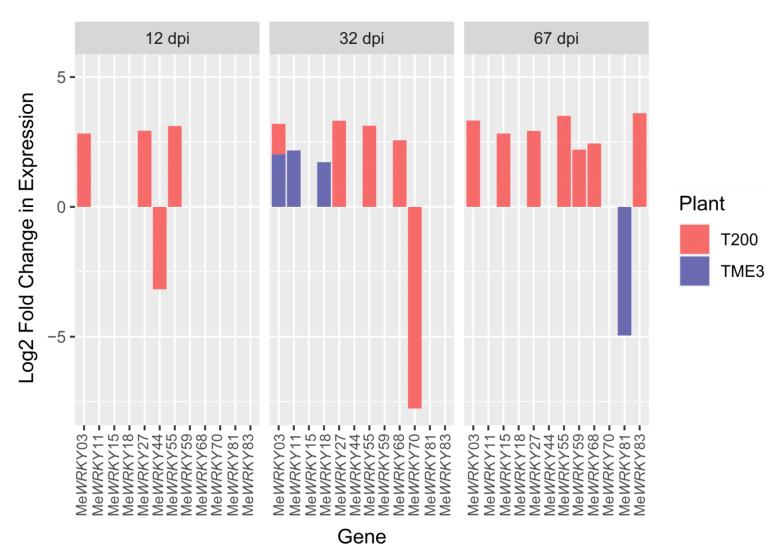
Log2 fold change in expression of *MeWRKY*s in T200 and TME3 at early (12 dpi), middle (32 dpi) and late (67 dpi) infection stages.

**Figure 5 viruses-13-01820-f005:**
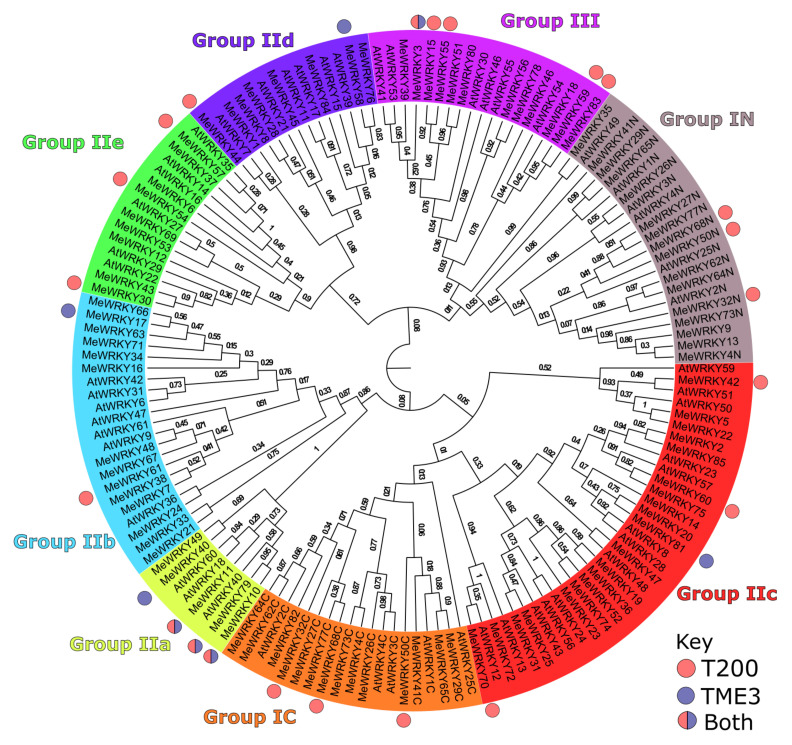
Phylogenetic tree of WRKY transcription factors from *Arabidopsis thaliana* (At) and cassava [*Manihot esculenta* (Me)] sorted into taxonomic groups based on amino acid sequences (bootstrap value of a 1000). Group I is characterized by two WRKY domains and two C2H2 zinc finger domains; Group II by one WRKY domain and a C2H2 zinc finger motif; and Group 3 has one WRKY domain and a C2HC zinc finger motif. Group 1 has been subdivided into two sub-groups (Group IC and Group IN) representing the alignment of the C and N termini of the protein, respectively. The tree (adapted from Wei et al., 2016) is annotated (dots) with the MeWRKYs identified as differentially expressed following SACMV infection.

**Figure 6 viruses-13-01820-f006:**
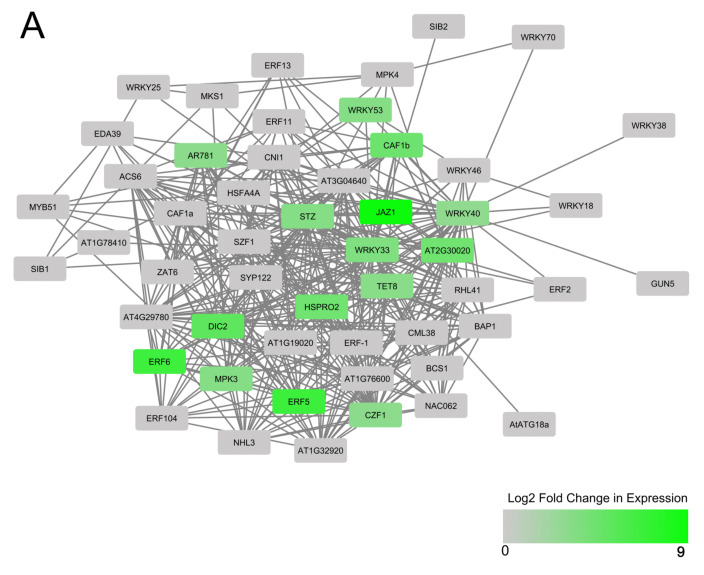
Predicted protein–protein interactions between AtWRKY homologs of differentially expressed MeWRKYs and their interacting partners in a central AtWRKY 33, 40, 53 and 70 protein-protein network (**A**) T200 and (**B**) TME3 at 32 days post SACMV inoculation. Networks were created using STRING v.11.

**Figure 7 viruses-13-01820-f007:**
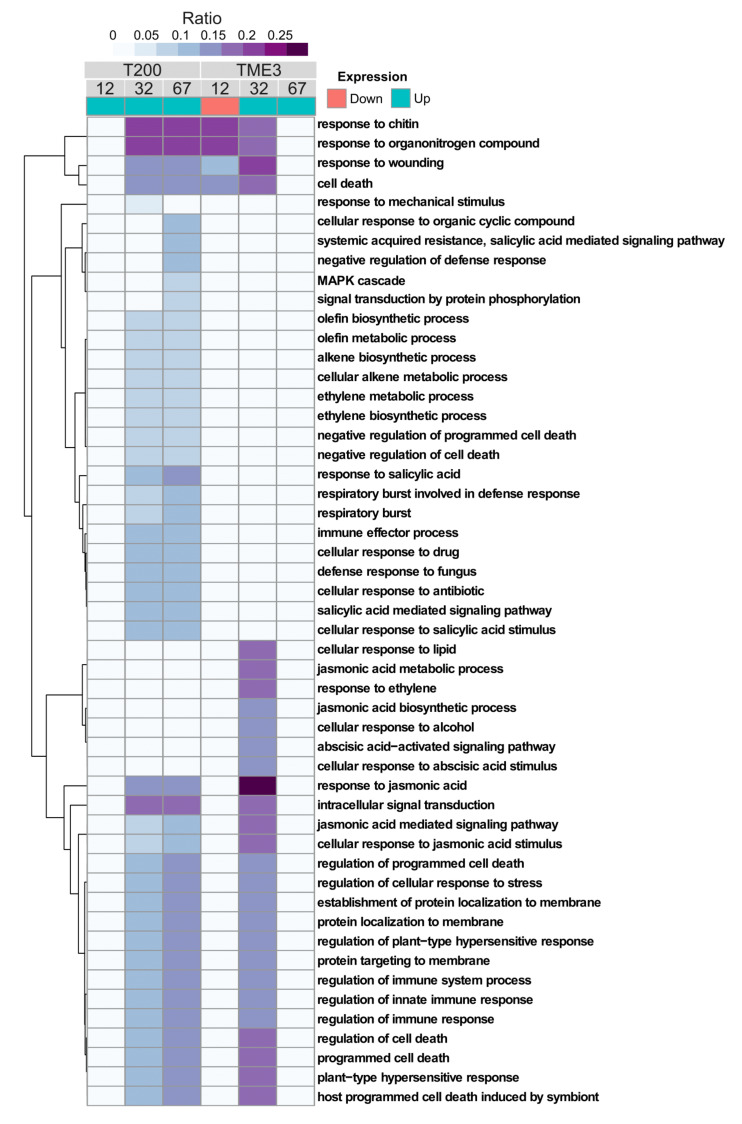
Heat map of significant (adjusted *p* value < 1 × 10^−8^) enriched GO terms from differentially expressed gene datasets in T200 and TME3 at 12, 32 and 67 days post *South African cassava mosaic virus* infection. The purple scale represents the ratio of regulated genes in each of the gene sets that fall into the enriched GO category. The blue bar blue represents upregulated genes and the red bar downregulated genes.

**Figure 8 viruses-13-01820-f008:**
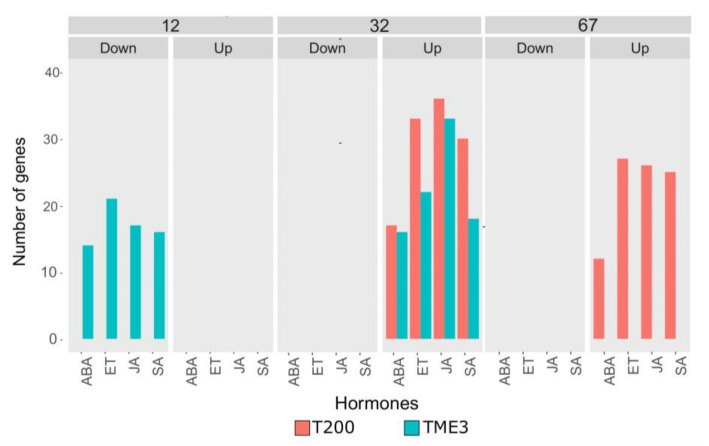
Number of upregulated or downregulated genes associated with enriched phytohormone GO terms (biological processes) in T200 and TME3 at 12, 32 and 67 days post inoculation with *South African cassava mosaic virus*.

**Figure 9 viruses-13-01820-f009:**
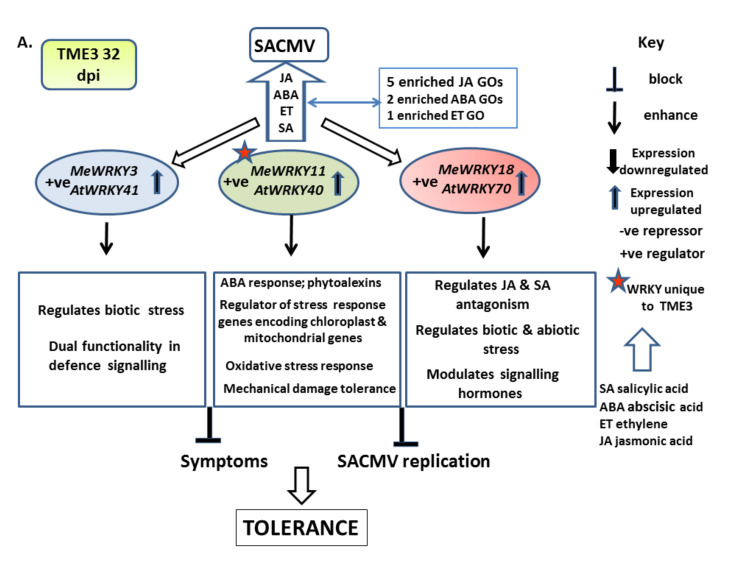
Proposed models of WRKY functions, hormones and downstream signaling of responses to SACMV in T200 and TME3 at 32 dpi. (**A**) TME3 exhibits milder symptoms compared to T200 in response to SACMV. This is associated with a number of upregulated hormone genes, JA, ABA, SA and ET, as a general response to infection. While a number of SA genes are upregulated, they are not significantly represented in the enriched GO (biological process) categories. In contrast to mock-inoculated plants, upregulation of *MeWRKY3*, *11* and *18* expression is observed. These transcription factors positively regulate hormone biosynthesis and signaling and biotic and oxidative stress functions. Differential expression of *MeWRKY11* is specifically associated with an ABA response, a hallmark of abiotic and biotic stress. These responses to WRKY expression contribute at a later stage to a lower viral load and symptom recovery observed at 67 dpi and tolerance to SACMV. (**B**) T200 exhibits severe symptoms in response to SACMV. This is associated with a number of upregulated hormones JA, ABA, SA and ET genes as a general response to infection. However, while a number of ABA genes are upregulated, they are not significantly represented in the enriched GO (biological process) categories. In contrast to mock-inoculated plants, upregulated expression of *MeWRKY3*, *5/55* and *70* is observed. AtWRKY12 is a known positive regulator of plant defense, and downregulation of expression of its *MeWRKY70* homolog contributes to severe leaf curl/chlorosis disease symptoms in T200. AtWRKY41 has been shown to regulate hormone signaling and biotic stress responses, and in cassava MeWRKY3, a homolog of AtWRKY41 plays a role in regulation of hormone signaling and biotic stress responses to SACMV. *Arabidopsis* NPR1 gene controls the onset of systemic acquired resistance (SAR)/immunity to a broad spectrum of pathogens, and AtWRKY53 is a known target of nonexpressor of pathogenesis-related gene (NPR1) during SAR in *Arabidopsis*. However, in the absence of SAR or the hypersensitive response (HR) in cassava, it is likely that SA-mediated responses of MeWRKY55, uniquely upregulated in T200, are associated with oxidative and other biotic stresses through cross-talk with JA and ET. The AtWRKY53 homolog of MeWRKY55 is a positive regulator of senescence suggesting that MeWRKY55 may play a role in other disease-associated metabolic perturbations associated with severe symptoms. (**C**) *Mitogen-activated phosphate kinase* (*MPK3*) is uniquely upregulated in T200 upon SACMV infection at 32 dpi (and at 67 dpi). Phosphorylation of MPK3 leads to upregulation of *MeWRKY27/68* (*AtWRKY33* homolog), a negative regulator of ABA and oxidative stress, which may contribute to high stress levels in T200 reflected by severe symptoms and high SACMV replication. While a number of ABA genes are upregulated in T200, there are no significantly enriched ABA biological processes observed in T200. This could also be a result of antagonism between SA, JA and ABA. *MeWRKY27/68* positively regulates an increase in ACC which is involved in ethylene biosynthesis. Ethylene while having many biological roles is widely hallmarked as the stress hormone. Ethylene responsive factors (ERF5 and 6) are upregulated; however, ERF5 does not appear to play a direct role in defense to SACMV in T200. ERF5 and 6 may play a general role in regulation of biotic stress.

**Table 1 viruses-13-01820-t001:** Log_2_ fold change in expression of *MeWRKY*s and the functions of their *Arabidopsis* homologs in susceptible T200 and tolerant TME3 landraces at 12, 32 and 67 days post infection (dpi) with *South African cassava mosaic virus*.

Landrace	Gene Name	At Homolog	12 dpi	32 dpi	67 dpi	Homolog Function	Reference	
TME3	*MeWRKY81*	*AtWRKY28*			−4.954	Positive regulator of JA signaling	Hu et al., 2013	
	*MeWRKY11*	*AtWRKY40*		2.17399		Transcriptional regulator of ABA genes	Pandey et al., 2010	
	*MeWRKY18/59*	*AtWRKY70*		1.72262		Regulates JA and SA antagonism	Li et al., 2004	
						Modulates secretion of signaling hormones	Li et al., 2006	
	*MeWRKY3*	*AtWRKY41*		2.02561		Confers positive resistance towards *P. syringae*	Hisagi et al., 2008	
T200	*MeWRKY44*	*AtWRKY07*	−3.1732			Negative regulator of defense signaling	Journot-Catalino et al., 2006	
	*MeWRKY70*	*AtWRKY12*		−7.7622		Positive regulator of defense against *P. carotovorum*	Kim et al., 2014	
	*MeWRKY27*	*AtWRKY33*	2.93383	3.32012	2.92518	Positive regulator of resistance against *B. cinerea*	Lai et al., 2011	
	*MeWRKY68*			2.56657	2.44239			
	*MeWRKY3*	*AtWRKY41*	2.82696	3.20307	3.32634	Confers positive resistance towards *P. syringae*	Hisagi et al., 2008	
	*MeWRKY15*	*AtWRKY53*			2.82623	Basal resistance against *P. syringae*	Hu et al., 2012	
	*MeWRKY55*		3.11448	3.12648	3.50639	Positive regulator of senescence	Miao et al., 2007	
	*MeWRKY18/59*	*AtWRKY70*			2.20366	Regulates JA and SA antagonism	Li et al., 2004	
	*MeWRKY83*				3.60985	Modulates secretion of signaling hormones	Li et al., 2006	

## Data Availability

Not applicable.

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
