# Peer review of "WRKY Transcription Factors in Cassava Contribute to Regulation of Tolerance and Susceptibility to Cassava Mosaic Disease through Stress Responses"

_viruses, 2021, doi:10.3390/v13091820_

Round 1
Reviewer 1 Report
This paper is an article titled by “WRKY Transcription Factors in Cassava Contribute to Regulation of Tolerance and Susceptibility to Cassava Mosaic Disease through Stress Responses” by Freeborough et al. using transcriptome analysis of a begomovirus, South African cassava mosaic virus (SACMV) infected cassava. After their previous studies (2018) this paper focused on WRKY transcription factors. The manuscript is not so highly remarkable, very confusing and rather wordy and not easy to understand for readers because their data and the data by other groups were mixed up. For examples Table 2 is not their results but just a list of previous papers. They should describe only what they did in the results. Title should reflect what they did actually. It is better to remove “through stress responses” because they did only SACMV inoculation. In the manuscript some genes were cited as AtXXX and others as MeXXX for cassava. There are no RT-qPCR data for MeWRKYs described here except MeWRKY40 (previous paper, 2018). From these points of view the manuscript needs vigorous clarification and improvements. From these the manuscript cannot be accepted for this Journal.
Reviewer 2 Report
I had a great opportunity to review research manuscript “WRKY Transcription Factors in Cassava Contribute to Regulation of Tolerance and Susceptibility to Cassava Mosaic Disease through Stress Responses” which is considered for publication in Viruses journal. Paper is well written, but it requires minor changes. Below I list a few remarks about the manuscript that, in my view, will improve it.
Lanes 132-140 : it looks like you gave more interest to the previous study than yours. It is recommended to summarize the previous study, highlight the link between this study and yours.
Lanes 140-153: Please reorganize this part by highlighting first the originality of your work, and then mentioning the different goals of your study (please remove the methods description in this section).
Lane 384: Please check the title numbering. It should be 3.4. instead of 3.3. Same remark for the subtitles of this section.
Lane 523: There is no need to spell out “CMD” as it has been already done in the introduction section. Please check this in all manuscript (in particular the discussion section). Abbreviations need to be spelled out one time in the abstract and another time in the manuscript (introduction or other) since they are independent parts.
The conclusions section needs to be improved according to the goals of the study. This section should first summarize the main findings of the work then present the perspectives. However, no discussion is needed in this part.
Remark for the whole manuscript: It is highly important to present your research in a precise and concise way. Please review the manuscript and remove some useless paragraphs if necessary.
Sincerely,
Reviewer 3 Report
1. Title
"WRKY Transcription Factors in Cassava Contribute to Regulation of
Tolerance and Susceptibility to Cassava Mosaic Disease through Stress
Responses"
Title is descriptive [good], and explains both what and how WRKY
transcription factors contribute. It uses the name Cassava (instead of
Manihot esculenta Crantz), but the name is commonly-used enough that
it should be obvious to the reader what it means.
---
---
2. Guessed key message from abstract
"Overall, WRKY, hormone and enriched biological processes results in
both landraces reflect oxidative and other biotic stress responses to
SACMV rather than defence."
This sentence would be better placed earlier in the abstract. The
abstract seems a bit too much like a results dump, rather than a
summary of results / conclusions.
---
---
3. Guessed key message from discussion
"Herein, we identified for the first time MeWRKYs involved in response
to cassava mosaic disease caused by South African cassava mosaic virus
in this orphan crop."
This sentence is buried a few lines down (line 540). The paper could
be improved by shifting the introductory sentences (prior to this
statement) to the introduction.
---
---
3a. Compare key messages in title, abstract, discussion
Key messages are roughly consistent, mentioning WRKY proteins. Stress
response is not emphasised in the first key message I could find in
the discussion. I have no suggestions about a better title; it looks good enough to me.
MeWRKY appears extensively in the abstract, but (from what I can tell)
the "Me" part is not explained at all in the manuscript. Presumably it
is an abbreviation for something like "Manihot esculenta WRKY
protein", but I can't find this full description anywhere in the
paper.
An ideal place for this explanation / description would be after
introducing WRKY proteins and discussing WRKY proteins from different
species (i.e around line 113).
---
---
4. Find the closest figure matching the dominant key message
[this is likely to have the most attention to detail]
Possibly Figure 8 (Heat map... in T200 and TME3 at 12, 32 and 67 days
post SACMV infection).
---
---
4a. How is normal / reference represented in this figure ?
Three time-point samples of T200 are compared to three time-point
samples of TME3, sampled at different times after infection. There are
no replicate samples. The displayed enriched GO terms are represented
on a scale (0 ~ 0.26), but the meaning of the scale is not clear from
reading the figure caption or image. The figure has additional "Up /
Down" annotation, but it is not clear what the Up and Down relate
to. I can't see anything obvious in the displayed graph to indicate
why the X12_TME3 "Down" annotation exists. The text where Figure 8 is
mentioned suggests that this annotation might relate to the "Up" or
"Down" nature of the gene sets, but I'm confused why that column is
similar in parts to other columns, but has a distinct annotation.
---
---
5. How are p-values used in the paper?
Significance thresholds are stated and used in combination with
fold-change statistics for filtering (good), but the reasoning behind
picking these thresholds is not discussed.
There is mention of an arbitrary threshold of "<1E-08" on line 440 as
the primary evidence for "no over-representation". An alternate
statistic is discussed on line 449 (namely "ratio of expressed genes
in TME3 at 32 dpi compared to [other ratios in other landraces]"), but
the relevance of this statistic is not obvious in the text where it is
discussed.
P-values do not feature prominently in the manuscript, and don't seem
to be the overriding reason for the stated conclusions (good).
The biological significance of the results is extensively discussed
(good), including a hypothesis [in text and figures] of the mode by
which tolerance and susceptibility occurs (Figure 10).
Normal ranges are not discussed. There does not seem to be any
replicate information that could be used to establish a normal range
of transcript expression.
There is some indication of adjusting p-value thresholds until results
are found (lines 861-866). My personal preference is to apply a loose
p-value threshold and rank by the magnitude of the difference
statistic, choosing as many genes (or GO terms, as it may be) that it
is reasonable to manually verify. Computer algorithms are good at
sensitively identifying interesting features, but poor at specifically
identifying features that make sense: use computers to find the
diamonds, and people to discard the dirt.
---
Other observations:
It's good to see that T200 is explained as a CMD-susceptible cassava
landrace the first time that it is mentioned.
WRKY is described extensively in the manuscript, but it's introduction
is buried a few sentences into the third paragraph of the introduction
(line 101):
"WRKY proteins contain the conserved DNA-binding domain, comprising
the WRKYGQK peptide and zinc-finger-like motifs, that binds to the
W-box cis-regulatory element of target genes...."
The paper could be improved by making this introductory sentence and
the subsequent descriptive sentences more prominent (e.g. by starting
a paragraph with these sentences at around line 91 with a separate
heading).
I find it curious, in light of this absence of introductory headings,
that the discussion begins with a heading for an introductory section.
In terms of English language editing, my main concern is that there is
a lot of text that can be overwhelming to process all at once. The
text itself reads well, but could be helped by being broken up into
more paragraphs.
I think that including components of Figure 10 as a visual abstract
for this manuscript would help with understanding of the manuscript.
Round 2
Reviewer 1 Report
The revised manuscript showed substantially improvements after incorporating the comments by reviewers. Most of them are understandable with some exceptions. However, the manuscript is not carefully written, especially in references. These should be carefully cheeked by following the guide line of this Journal. Some comments are described below.
- Considering the paper space, it would be better that Fig. 2 and 3 be combined at the same space (left and right instead of upper and down).
- 5: Letters are too small to notify each member at a glance. This fig. should be enlarged to the maximum size at this page (as much as possible from end to end of the same page).
- Line 1291: Evol Biol. should be Evol. Biol.
- Line 1311: Plant Journal should be Plant J.
- Line 1324: Plant cell, should be Plant Cell.
- Line 1328: Plant Molecular Biology should be Plant Mol. Biol.
- Line 1400: Front. Plant Sci. should be in italic and in lower case letters.
- Lines 1406 and 1422: PNAS should be Proc. Natl. Acad. Sci. USA.
- Line 1417: Plant physiol. should be Plant Physiol.
- References: Genus and species name is generally in italic. But most of them are not in italic in the manuscript. This type of correction is needed in many references here with some exceptions, depending upon reference paper.

Author Response
Responses to Reviewer 1
The revised manuscript showed substantially improvements after incorporating the comments by reviewers. Most of them are understandable with some exceptions. However, the manuscript is not carefully written, especially in references. These should be carefully cheeked by following the guide line of this Journal. Some comments are described below.
- Considering the paper space, it would be better that Fig. 2 and 3 be combined at the same space (left and right instead of upper and down).
Author Response: This has been done. I have left the old figures and inserted the new one with figures 2 and 3 side by side.
- Fig. 5: Letters are too small to notify each member at a glance. This fig. should be enlarged to the maximum size at this page (as much as possible from end to end of the same page).
Author Response: The figure was increased in size to fit the maximum width of the page. Again I have kept old figure and inserted new figure.
Reference corrections
- Line 1291: Evol Biol. should be Evol. Biol.
- Line 1311: Plant Journal should be Plant J.
- Line 1324: Plant cell, should be Plant Cell.
- Line 1328: Plant Molecular Biology should be Plant Mol. Biol.
- Line 1400: Front. Plant Sci. should be in italic and in lower case letters. [I checked and it is Front. Plant Sci. not in lower case].
- Lines 1406 and 1422: PNAS should be Proc. Natl. Acad. Sci. USA.
- Line 1417: Plant physiol. should be Plant Physiol.
Response: All of these above have been corrected except Front. Plant Sci.
- References: Genus and species name is generally in italic. But most of them are not in italic in the manuscript. This type of correction is needed in many references here with some exceptions, depending upon reference paper.
Response: So some journal articles refer to the virus name generically and some as a species in the article title. As an example I checked on PubMed (ftp://ftp.ncbi.nih.gov/pubmed/J_Medline.txt) and this article for example did not use the species name in italics in this paper: Tabein S, Miozzi L, Matić S, Accotto GP, Noris E. No Evidence for Seed Transmission of Tomato Yellow Leaf Curl Sardinia Virus in Tomato. Cells. 2021 Jul 2;10(7):1673. doi: 10.3390/cells10071673. PMID: 34359841; PMCID: PMC8306144.
So when quoting in this manuscript do you put the virus name or genus in italics in the reference list if this was not in italics in the original article?? Decision: I have changed all virus species, plant species and genera to italics in the reference list in this manuscript.
